# Autophagy Promotes Enrichment of Raft Components within Extracellular Vesicles Secreted by Human 2FTGH Cells

**DOI:** 10.3390/ijms25116175

**Published:** 2024-06-04

**Authors:** Valeria Manganelli, Luciana Dini, Stefano Tacconi, Simone Dinarelli, Antonella Capozzi, Gloria Riitano, Serena Recalchi, Tuba Rana Caglar, Federica Fratini, Roberta Misasi, Maurizio Sorice, Tina Garofalo

**Affiliations:** 1Department of Experimental Medicine, “Sapienza” University of Rome, 00161 Rome, Italy; valeria.manganelli@uniroma1.it (V.M.); antonella.capozzi@uniroma1.it (A.C.); gloria.riitano@uniroma1.it (G.R.); serena.recalchi@uniroma1.it (S.R.); tubarana.caglar@uniroma1.it (T.R.C.); roberta.misasi@uniroma1.it (R.M.); tina.garofalo@uniroma1.it (T.G.); 2Department of Biology and Biotechnology C. Darwin, “Sapienza” University of Rome, 00185 Rome, Italy; luciana.dini@uniroma1.it; 3CarMeN Laboratory, INSERM 1060-INRAE 1397, Department of Human Nutrition, Lyon Sud Hospital, University of Lyon, 69310 Lyon, France; stefano.tacconi@inrae.fr; 4Institute for the Structure of Matter (ISM), National Research Council (CNR), Via del Fosso del Cavaliere 100, 00133 Rome, Italy; simone.dinarelli@ism.cnr.it; 5Proteomics Core Facility, Istituto Superiore di Sanità (ISS), 00161 Rome, Italy; federica.fratini@iss.it

**Keywords:** autophagy, extracellular vesicles, exosomes, lipid rafts, ERLIN1

## Abstract

Autophagy plays a key role in removing protein aggregates and damaged organelles. In addition to its conventional degradative functions, autophagy machinery contributes to the release of cytosolic proteins through an unconventional secretion pathway. In this research, we analyzed autophagy-induced extracellular vesicles (EVs) in HT1080-derived human fibrosarcoma 2FTGH cells using transmission electron microscopy and atomic force microscopy (AFM). We preliminary observed that autophagy induces the formation of a subset of large heterogeneous intracellular vesicular structures. Moreover, AFM showed that autophagy triggering led to a more visible smooth cell surface with a reduced amount of plasma membrane protrusions. Next, we characterized EVs secreted by cells following autophagy induction, demonstrating that cells release both plasma membrane-derived microvesicles and exosomes. A self-forming iodixanol gradient was performed for cell subfractionation. Western blot analysis showed that endogenous LC3-II co-fractionated with CD63 and CD81. Then, we analyzed whether raft components are enriched within EV cargoes following autophagy triggering. We observed that the raft marker GD3 and ER marker ERLIN1 co-fractionated with LC3-II; dual staining by immunogold electron microscopy and coimmunoprecipitation revealed GD3-LC3-II association, indicating that autophagy promotes enrichment of raft components within EVs. Introducing a new brick in the crosstalk between autophagy and the endolysosomal system may have important implications for the knowledge of pathogenic mechanisms, suggesting alternative raft target therapies in diseases in which the generation of EV is active.

## 1. Introduction

The endomembrane system (EMS) of the mammalian cells, which includes the endoplasmic reticulum (ER), Golgi apparatus, plasma membrane (PM), lysosomes, and vesicular cargos (endosomes and autophagosomes), plays an essential role in ensuring cell homeostasis because of its capacity to segregate biochemical reactions into different compartments [1,2]. Indeed, EMS is enriched in a variety of molecular pathways able to regulate specific processes, such as protein and lipid synthesis, degradation, and autophagy inside cells and among cells.

Emerging evidence from the last few decades has identified two secretory (vesicular) pathways of EMS: autophagy, an intracellular pathway during which cytosolic proteins and organelles are captured into autophagosomes and finally degraded through fusion with lysosomes [3,4], and exosome biogenesis, well known for the formation and release of vesicles into the extracellular space, that is, the small extracellular vesicles (sEVs) [5].

Both autophagy and exosome biogenesis are tightly regulated processes that display synergies in degradation, recycling, and secretion [2]. In particular, autophagy plays a crucial role in selectively removing protein aggregates and dysfunctional organelles, while molecular constituents, including amino acids, lipids, and sugars derived from degradation, contribute to cell survival, especially under nutrient deprivation. Upon induction of autophagy, a complex of autophagy-related genes (Atg), including Atg5, Atg12, and Atg16L1, terminates an enzymatic cascade, culminating in the covalent linkage of ATG8 [microtubule-associated protein 1 light chain 3B (MAP1LC3B) or LC3B in mammals] to phosphatidylethanolamine (PE) to form LC3-II [6]. The generation of LC3-II is canonically involved in the expansion of the phagophore and its closure to form a double-membraned autophagosome. The final step of autophagy takes place when autophagosomes fuse with lysosomes to generate autolysosomes, and cargos are degraded by acid hydrolases while their components are recycled.

Beyond its well-known functions, autophagy can regulate the secretion of cytosolic proteins through an unconventional secretion pathway. In fact, autophagy has also been shown to facilitate the movement of membrane proteins to the plasma membrane [7,8], demonstrating that it affects intercellular communication. In this regard, it has been reported that the LC3B-positive carrier plays a role in moving the cytokine interleukin 1β (IL-1β) from the cytosol towards the plasma membrane through a process described by many authors as “secretory autophagy” [9,10]. Recent evidence reported that, in a pre-lysosomal step, autophagosomes may undergo fusion with multivesicular endosomes, containing several intraluminal vesicles (ILVs), precursors of exosomes [5], to form hybrid vesicles, termed amphisomes, which at the final step combine with lysosomes for hydrolysis of cargos. Alternatively, amphisomes can fuse with the PM, resulting in the release of exosomes into the extracellular space [11,12]. Recently, amphisome biogenesis has gained increasing interest. Several key molecules have been indicated to drive amphisome generation, such as soluble N-ethylmaleimide-sensitive factor activating protein receptors (SNAREs), Rab-GTPases tethering complexes, and endosomal sorting complexes required for transport (ESCRT) [13,14]. However, the colocalization of exosome markers, including CD63 and CD81, with autophagosomal proteins p62 and LC3 supports the sharing of additional molecular pathway(s) between autophagy and exosome biogenesis in generating amphisomes [15]. Many studies in this field have addressed the fate and role of these vesicles in both intracellular and intercellular communication. Their ability to cooperate with autophagy flux for preserving cellular homeostasis has recently been reported, and some autophagy-related proteins have been investigated, as revealed for ATG9 in the regulation of intraluminal vesicle formation of endosomes before amphisome biogenesis [16]. The interplay between autophagy and exosome release in tumor progression has elicited interest in the last few years. Cancer cells release more exosomes, which contribute to many aspects of tumor malignity, including proliferation, invasion, metastasis, angiogenesis, and immunosuppression [17,18,19]. Recently, an autophagy-independent function of the ATG5/ATG16L1 complex on exosome biogenesis and metastasis has been suggested [20]. Guo et al. identified a novel mechanism for multivesicular bodies (MVBs) and exosome biogenesis with a significant impact on migration, invasion, and metastasis in models of breast cancer. Depletion of ATG5 or ATG16L1 significantly decreases exosome release and attenuates the enrichment of LC3-II at the exosomal level, supporting the role of “ATG5-dependent secretion”.

Different molecular components, including lipids, tetraspanins, and microdomains localized on distinct intracellular membrane compartments, participate in MVB membrane inward budding and exosome sorting both in physiological [21,22] and pathological processes [23]. Interestingly, these studies showed a high enrichment in lipid species within exosomes, mainly in glycosphingolipids, sphingomyelin, cholesterol, and phosphatidylserine [24].

In this regard, during EV biogenesis, proteolipid proteins (PLP) are recruited into ILVs in the absence of the ESCRT machinery subunits by lipid rafts. These regions of cell membranes are specific microdomains enriched in sphingolipids, from which ceramides are generated by the activity of sphingomyelinases. Ceramide stimulates the assembly of these microdomains and prompts ILV generation. Thus, ceramide seems to play a pivotal role in intraendosomal membrane transport and the formation of another population of ILVs that is not destined for transport towards lysosomes but is secreted as a class of exosomes [25,26]. Interestingly, the addition of ceramide to multiple myeloma cells promoted exosome secretion [27].

In a previous study, we analyzed the role of raft components in the regulation of the autophagic process using HT1080-derived human fibrosarcoma 2FTGH (2F) cells [28]. Thus, in the present research, we used the same tumorigenic cell line to investigate the enrichment of raft components within EVs induced by autophagy using morphological and biochemical approaches. Of note, this cell line has been previously used as an in vitro model for autophagy activation [29,30]. In particular, we analyzed whether raft components are enriched within EV cargoes released from 2FTGH cells into the extracellular milieu following autophagy triggering.

## 2. Results

### 2.1. Autophagy Induces the Formation of Intracellular Enlarged Organelles in 2FTGH Cells

It is well known that in a pre-lysosomal step, the autophagosome may undergo fusion with MVBs to form a hybrid intracellular organelle, that is, the amphisome. Thus, in this study, with the aim of investigating whether induction of autophagy promotes the formation of enlarge organelles, human fibrosarcoma 2FTGH cells were incubated in an amino acid and serum-deprived medium (HBSS) for 4 h and analyzed first by transmission electron microscopy (TEM). As shown in Figure 1A (left panel), in untreated cells, a subset of heterogeneous intracellular MVB-like structures in proximity to the plasma membrane was detectable (Figure 1A, right panel, red arrows in red box). Interestingly, in HBSS-starved 2FTGH cells, the formation of intracellular enlarged organelles containing small internal vesicles, that is, ILV, was evident, as observed in Figure 1A (see red arrow in red box).

Next, to better analyze the cell surface and budding vesicle distribution of untreated and HBSS-treated cells, atomic force microscopy (AFM) analysis was employed. As shown in the left panel of Figure 1B, the analysis by AFM showed that the cell surface of untreated 2FTGH cells was rough and characterized by the abundant presence of protrusions and budding vesicles. Notably, HBSS treatment led to a more visible smooth cell surface with a reduced amount of cell surface protrusions, as shown in the right panel in Figure 1B (see arrows). Western blot analysis was performed to check autophagy activation both in control and 4h HBSS-treated 2FTGH cells by using anti-MAP1LC3/LC3 (microtubule-associated protein 1 light chain 3) or anti-p62/SQSTM1 (sequestosome 1) antibodies (Figure 1C). The analysis revealed an increase in LC3-II and a decrease in LC3-I after 4h of cell starvation. This finding suggests an accumulation of autophagosomes. As expected, we observed a significant decrease in p62/SQSTM1 together with the increase in LC3-II after cell starvation (Figure 1C, left panel), as confirmed by densitometric analysis (Figure 1C, right panel, bar graphs). In addition, to verify the efficiency of HBSS treatment in inducing early autophagy, we also investigated the expression levels of ATG5 and ATG7 markers involved in phagophore expansion. Western blot analysis revealed an increase in both proteins in HBSS-treated 2FTGH with respect to control cells, as confirmed by densitometric analysis (Figure 1C, right panel, bar graphs), demonstrating the integrity of the autophagic process in these cells. Thus, these results indicate that the activation of the autophagic process is accompanied by heterogeneous MVB-like structure generation, as revealed by TEM, according to the role of autophagy in triggering the formation of MVs [2].

### 2.2. Autophagy Induces a Differential Budding Profile in 2FTGH Cells, as Revealed by AFM Topographical Mapping

Next, with the aim of better clarifying the presence of a differential budding profile in distinct cellular regions, including nuclear (Nu; 6 × 6 μm^2^) and peripheral areas (perinuclear-to-cellular edge Pn/CE; 6 × 6 μm^2^), AFM topographical analysis, and 3D reconstructions were performed in untreated and HBSS-treated cells. As shown in Figure 2A, for each image, a zoom on one of these regions has been highlighted and represented as 3D views to better visualize the spatial arrangement of the membrane. As expected, the exposure to particular stimuli induced localized changes in vesicles budding due to the finely regulated process of EV release, in which membrane rearrangements take place in specific regions of the cell in response to environmental conditions. Thus, in these selected regions, Nu and Pn/CE, waviness, a simplified profile of the entire cell of the selected area, as well as membrane roughness (Rq), a key parameter that is directly related to the cellular surface arrangement and topography at the nanoscale level, were measured (Figure 2B). Rq represents a direct estimation of the good attachment of the membrane to the underlying cytoskeleton and can be used to evaluate the general cellular status [31,32]. An example of roughness and waviness over the entire cell is shown in Figure 2B for one representative treated cell compared with a control one. The comparative analysis clearly revealed that the values of the linear roughness (black lines) were quite different between the nuclear and perinuclear regions (separated by dashed vertical gray lines). To provide a quantitative analysis, for each cell, the roughness values in 2 or 3 (depending on the total dimension of the cell itself) not-overlapping 6 × 6 μm areas onto the Nu and Pn/CE regions, obtaining a representative average value of the roughness of the region, were computed and reported in Figure 2C. The analysis of the only nuclear region revealed that there was a statistically significant difference between the untreated and treated cells average roughness values. However, a highly statistically significant difference was found between the Nu and Pn/CE in HBSS-treated cells [33], with a decrease in Rq values in the peripheral regions of the cells.

### 2.3. Characterization of Small and Large EVs Secreted by 2FTGH Cells following Autophagy Induction

In parallel experiments, untreated and HBSS-treated cells were incubated in EV-depleted medium for 24 h. Conditioned medium from each sample was collected and subjected to standard differential centrifugation, as described by Jeppesen et al. [34]. Different pelleted materials were recovered: by centrifugation at 15,000× *g* for large EVs (P15) and by ultracentrifugation at 120,000× *g* for small EVs (P120). Large EVs and small EVs were then compared. Importantly, under these experimental conditions, less than 10% cell death was observed, and the amount of material recovered in the pellet did not positively correlate with cell death revealing that the large EVs analyzed are not apoptotic or necrotic cell fragments. When analyzed using TEM (see the representative TEM images in Figure 3A), P15 pellets showed EVs in the majority larger than 150 nm in diameter with a broad size distribution (Figure 3B). In contrast, the majority of the P120 EVs revealed a diameter of 40–150 nm (small EVs, consistent with the size of exosomes, Figure 3B). The size distribution of the small EVs was different in HBSS-treated cells as compared to untreated cells; in particular, the small EVs from HBSS-treated cells were larger as compared to EVs from untreated cells, with a heterogeneous size distribution (Figure 3B). Next, to better characterize the mixed EV populations, Western blot analysis of P15 and P120 pellets was performed. Interestingly, the most common exosome markers, including tetraspanin CD63 (Appendix A), tetraspanin CD81, and ALG-2 interacting protein X (ALIX) (Figure 3C), were highly enriched in the P120 pellet containing the small EVs as compared to whole cell lysates. As expected, Annexin A1 (ANXA1), a specific marker for plasma membrane-derived EVs, as well as GM130, a specific intracellular marker, were virtually absent in the P120 pellet. On the contrary, we found the presence of ANXA1 in the P15 pellet, containing the large EVs (Figure 3C).

### 2.4. HBSS Treatment Increases the Release of Small EVs from 2FTGH Cells

To quantify the release of small EVs, 2FTGH cells were treated with HSSS for 4 h, and EVs released during the next 24 h were then isolated from the conditioned media by ultracentrifugation (120,000× *g*). We preliminary measured the concentration of small EVs from untreated and HBSS-treated cells by TEM, revealing an increase in small EVs in HBSS-treated cells with respect to untreated cells (*p* < 0.05) (Table 1). In parallel experiments, small EVs were resuspended in PBS and quantified by nanoparticle tracking analysis (NTA). As shown in Figure 4, HBSS treatment induced an increase in particles/mL levels with respect to control. To monitor whether functional inhibition of autophagosome/lysosome fusion triggers EVs release, we tested the effects of HBSS on EV release in combination with the lysosomal inhibitor bafilomycin A1 (BafA1). Then, EVs were collected and analyzed by NTA, as described above. We found that Baf1 treatment increased EV release in pretreated HBSS cells with respect to HBSS-treated cells without BafA1 treatment (Figure 4).

### 2.5. Analysis of Small EVs by 6–30% Iodixanol Gradient Density

To purify small EVs from non-vesicular (NV) components, we employed high-resolution iodixanol gradients of crude pellets of small EVs (P120) isolated from both untreated and HBSS-treated 2FTGH cells. Next, the EVs were subjected to density gradient ultracentrifugation to separate membrane-enclosed vesicles according to their floatation speed and equilibrium density [34]. Twelve fractions were collected from the top to the bottom of the gradient tube. Thus, we checked the presence of the LC3-II marker in the 12 fractions recovered from the iodixanol gradient obtained either from untreated or HBSS-treated 2FTGH cells. As known [35], autophagy induction leads to an increased processing of LC3 with the conversion of soluble LC3-I to lipid-membrane-bound LC3-II on the autophagosome; the accumulation of lipidated LC3-II can be directly assessed using Western blot analysis. Remarkably, we found that endogenous LC3-II co-fractionated with both CD63 and CD81 (Figure 5A) in characteristic low-density fractions (3–5), indicative of small EVs enriched in exosomes. Otherwise, LC3-II appeared virtually absent in high-density fractions (8–12) enriched in non-vesicular components (NV). After treatment with HBSS, we observed a significant increase in the LC3-II level in fractions 3–5. Recently, we demonstrated that mitochondria-associated membranes (MAMs), identified as lipid raft-like subdomains located on ER membranes in close contact with mitochondria, play a crucial role in the early step in the formation of autophagosomes [4,36]. Moreover, in these areas, we identified two main markers, namely, the ganglioside GD3 and the protein ERLIN1 [28,37]. Thus, in this research, we verified whether lipid raft components, that is, GD3 and/or ERLIN1, might also be present as components of the small EVs secreted by 2FTGH cells following autophagy triggering. To this purpose, we first analyzed the ERLIN1 distribution in the same 12 fractions. In untreated cells, we found ERLIN1 essentially localized in low-density fractions (3–5), similarly to what was observed for LC3-II distribution. After treatment with HBSS, we observed a significant increase in the ERLIN1 content in fractions 3–5 (Figure 5A), as also confirmed by densitometric analysis (Figure 5B). TEM analysis revealed that floatation into the iodixanol gradient allows separation of subtypes of EVs displaying different buoyant densities and sizes, with small vesicles highly enriched in the light fraction corresponding to fraction 5 of HBSS-treated cells, revealing the presence of vesicles consistent with the presence of amphisomes-like (Figure 5C). Otherwise, as expected, in the high-density NV corresponding to fraction 11 from both untreated and HBSS-treated cells, vesicles appeared virtually absent (Figure 5C).

### 2.6. Autophagy Induction Promotes LC3-II and GD3 Association within Small EVs

Next, we investigated the presence of GD3 in the iodixanol gradient fractions of both untreated and treated cells. Of interest, GD3 co-fractionated with both LC3-II and ERLIN1 in fractions 3–5 obtained from cells under autophagy-stimulating conditions. After treatment with HBSS, we observed a significant increase in GD3 content in fractions 3–5, as revealed by densitometric analysis (Figure 6A). Immunogold electron microscopy was also performed to better assess the presence of ganglioside GD3 and its interaction with LC3 on small EVs from untreated and HBSS-treated cells. As shown in Figure 6B, under starvation conditions, an increase in gold particles labeling GD3 was observed. Using gold particles of different sizes to detect GD3 and LC3-II, we observed the presence of both GD3 and LC3 in the same vesicle, clearly indicating an association between the two molecules (Figure 6B and Appendix A). This association was strongly supported by the observation that GD3 coimmunoprecipitated with LC3-II, mainly after autophagy stimulation, as revealed by densitometric analysis (Figure 6C). These findings suggest the presence of both GD3 and LC3-II as components of a multimolecular complex within EVs of autophagic origin, which could also include other crucial proteins for autophagosome biogenesis and maturation, namely, ERLIN1 [28].

## 3. Discussion

In previous studies, we demonstrated the involvement of lipid rafts localized at the ER-mitochondria-associated membranes during autophagosome formation [4]. At these sites, crucial reactions can be catalyzed, profoundly influencing the regulation of intracellular trafficking and sorting, cholesterol homeostasis, and cell fate. Mitochondria-associated membranes (MAMs) have been classified as critical “hubs” in the regulation of apoptosis, autophagy, and tumor growth [4,28,38,39,40]. Since ER-membrane contact sites may be considered key platforms for the biogenesis of EVs [26], in this study, we characterized the molecular composition of autophagy-induced EVs, focusing on the presence of raft components. As a cellular model, we used untransfected control 2FTGH human fibrosarcoma cells, which represent a good model to study enrichment of raft components within EVs promoted by autophagy, since in previous studies, the autophagy process was analyzed in detail in these cells [29,30] and, mainly, we studied in this cell line the role of raft components in autophagosome formation [28,38]. In particular, we planned to use a tumorigenic cell line since EVs in tumor cells are known to have a significant impact during cancer progression [20].

In a first set of experiments, by using methodologies for EV morphological analysis, including TEM and AFM, we observed that autophagy induces the formation of a subset of large heterogeneous intracellular vesicular structures, suggestive of intracellular enlarged organelles in human fibrosarcoma 2FTGH cells, containing small ILV. Thus, these data suggest that autophagy triggering leads to an up-regulation in fusion between MVBs and autophagosomes and, consequently, the generation of enlarged hybrid organelles, that is, amphisomes. In addition, AFM analysis shows that the cell surface of untreated cells is rough and characterized by the abundant presence of protrusions, most of which end with a budding vesicle. Conversely, HBSS treatment led to a more visible smooth cell surface with a reduced amount of cell surface protrusions, suggesting vesicle release to the extracellular environment. EVs generated from the endosomal system have been shown to regulate the autophagic process, since biochemical studies support the evidence that autophagy shares with EVs autophagy-related proteins and key molecules for EVs biogenesis and secretion pathways [2,41]. Although in most cases, autophagy activation is associated with the lysosomal degradation of EVs, emerging studies showed that in response to upstream signals leading to inactivation of protein kinase mammalian target of rapamycin (mTORC1), rapamycin treatment [37,42], growth factor deprivation, or exposure to physical stress [43], autophagy can regulate the rate of EVs release as well as their composition.

Next, we characterized EVs secreted by cells following autophagy induction, demonstrating that human 2FTGH cells release a large range of EVs, including plasma membrane-derived microvesicles and exosomes, corresponding to large and small EVs, respectively. In addition, the size distribution of large EVs and small EVs was different in HBSS-treated cells, since large EVs showed a broader size distribution besides a lower mean diameter. Opposite results were found for small EVs of HBSS-treated cells, whose mean diameter increased, as well as the size distribution. In addition, NTA analysis revealed an increased release of small EVs expressed as particles/mL after cell starvation with respect to untreated cells. To monitor whether functional inhibition of autophagosome/lysosome fusion alters the rate of EV release, we tested the effects of HBSS in combination with the lysosomal inhibitor BafA1. As previously reported [44], we found that BafA1 treatment increased EV release from 2FTGH human fibrosarcoma cells.

A self-forming iodixanol (OptiPrep) gradient was performed for cell subfractionation. To elucidate the contribution of the inner autophagosome membrane to exosome generation from autophagy-induced cells, we checked the presence of the LC3-II marker in the same fractions. In a pre-lysosomal step, the autophagosome may undergo fusion with MVBs to form a hybrid organelle, termed amphisome, which finally combines with lysosomes for hydrolysis and degradation of cargos; alternatively, they fuse with the plasma membrane for the release of EVs in the extracellular space [5]. Remarkably, Western blot analysis of the 12 fractions recovered from the iodixanol gradient obtained either from untreated or HBSS-treated cells showed that endogenous LC3-II co-fractionated both with CD63 and CD81, the well-defined exosome markers, at characteristic buoyant densities, with a slightly broader distribution into less dense fractions.

Recently, we demonstrated that MAMs, identified as lipid raft subdomains located on ER membranes in close contact with mitochondria, play a crucial role in the early step in the formation of autophagosomes [4]. In these areas, we identified a strict interaction between GD3 (the raft marker) and ERLIN1 (an ER marker) responsible for modulating starvation-induced association of core complex molecules at MAMs level and, as a consequence, autophagosome biogenesis and maturation. Thus, a relevant finding of the present study is the demonstration that both GD3 and ERLIN1 co-fractionated with LC3-II in the same fractions. Moreover, dual staining by immunogold electron microscopy revealed evident colocalization areas between GD3 and LC3-II. These findings, together with coimmunoprecipitation experiments, indicate that autophagy promotes the enrichment of raft components within EVs secreted by the human fibrosarcoma cell line. It confirms and extends previous observations demonstrating that secretory autophagosomes originate from omegasome-like cell components, potentially from ER, and fuse with the plasma membrane for the secretion of their content. In addition, ER subdomains interacting with mitochondria (MAMs) contain raft components, which interact with the core initiator proteins AMBRA1 and WIPI-1 [4]. The only limitation of this study could be the use of a single tumorigenic cell line. Further studies will be needed to evaluate whether it is a general phenomenon not exclusive to 2FTGH cells.

Both exosomes and microvesicles are characterized by a lipid bilayer that contains intraparticle proteins and metabolites, which support transmembrane and surface-bound proteins. In particular, some groups have characterized the membranes of exosomes as enriched with the same lipid compositions as lipid raft-like microdomains [24,45,46], suggesting that they might be directly formed from lipid rafts. In particular, increased levels of free cholesterol and sphingomyelin further indicate that the lipid profile of exosomes is similar to that of lipid rafts, but it is unclear whether this lipid composition has functional significance or is simply related to the site of vesicle generation, where the predominant composition is that of a lipid raft. However, lipids may be important for the sorting of specific proteins into exosomes. Interestingly, exosomes have been shown to be enriched in cholesterol, sphingomyelin, and glycosphingolipids compared to their parent cells [47,48]. This suggests that exosomal membranes may contain lipid rafts, membrane subdomains enriched in cholesterol, and glycosphingolipids that play important roles in signaling and sorting [49,50]. In fact, one of the first studies on the role of lipids in exosome release showed that lyn, flotillin-1, and stomatin are released to the extracellular medium via their association with lipid domains in the exosomal membrane [49,50]. In addition, sphingosine 1-phosphate (SP1) has been shown to regulate cargo (such as CD63, CD81, and flotillin) sorting into exosomes via inhibitory G protein (Gi)-coupled S1P receptors located on MVB membranes [51]. In this regard, the use of drugs targeting lipid rafts, such as cyclodextrins or statins, may be a useful tool for personalized medicine in selected patients. In recent years, EVs have gained significant attention for their pivotal roles in various diseases. Introducing a new brick in the crosstalk between autophagy and the endolysosomal system may have important implications for the knowledge of pathogenic mechanisms and the development of focused therapeutic strategies [52]. In neurodegenerative diseases, enhanced exocytosis of amphisomes can mitigate the toxicity induced by the aberrant accumulation of protein aggregates/amyloid [53,54]. Moreover, autophagy and EV secretion are essential cellular processes with intricate roles in cancer development and progression, since exosomes are implicated in intercellular communication and tumor microenvironment modulation [55]. Thus, manipulation of amphisome secretion through membrane trafficking pathway(s) would be a promising strategy for therapeutic approaches in disease prevention.

## 4. Materials and Methods

### 4.1. Cells and Autophagy Induction

Human 2FTGH (2F) cells (provided by ECACC, 12021508) derived from HT1080 human sarcoma cells [56] were cultured in Dulbecco’s Modified Eagle Medium (DMEM; Sigma Aldrich, St. Louis, MO, USA, D5796), supplemented with 10% fetal calf serum (FCS; Aurogene, AU-S1810, Rome, Italy), 100 units/mL penicillin, and 10 mg/mL streptomycin (Aurogene, Rome, Italy, AU-L0022), at 37 °C in a humified CO_2_ atmosphere. Cells were treated for autophagy induction under conditions of nutrient deprivation with Hanks’ Balanced Salt Solution (HBSS; Sigma Aldrich, H9269) for 4 h at 37 °C. The optimal incubation time with HBSS was determined based on preliminary experiments. To inhibit autophagic flux, 1 μM Bafilomycin A1 (Baf A1) (Sigma-Aldrich B1793) was added for 1h after HBSS treatment [28]; then, cells were incubated in EV-depleted medium for 24 h. Following treatment, cells were collected and processed for the experimental procedures indicated below.

### 4.2. Analysis of Autophagy

Cells, untreated or treated with HBSS for 4 h at 37 °C, were lysed in lysis buffer, containing 1% Triton X-100 (Bio-Rad, Hercules, CA, USA, 1610407), 10 mM Tris-HCl, pH 7.5, 150 mM NaCl, 5 mM EDTA, 1 mM Na_3_VO_4_ (Sigma Aldrich, 450243), and 75 U of aprotinin (Sigma Aldrich, A1153) for 20 min at 4 °C. The lysate was centrifuged for 5 min at 1300× *g* to eliminate nuclei and large cellular debris. After evaluation of the protein concentration by Bradford Dye Reagent assay (Bio-Rad, 500–0006), the lysate was subsequently subjected to 15% sodium-dodecyl sulfate polyacrylamide gel electrophoresis (SDS-PAGE). The proteins were transferred electrophoretically onto polyvinylidene difluoride (PVDF) membranes (Bio-Rad, 162–0177). Membranes were blocked with 5% nonfat dried milk (Santa Cruz Biotechnology, Dallas, TX, USA, sc-2325) in TBS (Bio-Rad, 1706435), containing 0.05% Tween 20 (Bio-Rad, 1706531) and probed with rabbit anti-LC3 antibodies (Novus Biologicals, Centennial, CO, USA, NB100-2331) or rabbit anti-p62/SQSTM1 mAb (Cell Signaling Technology, Danvers, MA, USA, 8025), or rabbit anti-Atg5 pAb (MBL, Woburn, MA, USA, pm050), or rabbit anti-Atg7 (D12B11) mAb (Cell Signaling Technology, 8558), or mouse anti-β-actin antibodies (Sigma-Aldrich, A5316). Then, membranes were incubated with horseradish peroxidase (HRP)-conjugated anti-rabbit IgG (Sigma Aldrich, A1949) or anti-mouse IgG (Sigma Aldrich, A9044) antibodies to visualize the bound antibodies, and the immunoreactivity was evaluated by a chemiluminescence reaction with the ECL western detection system (Amersham, Buckinghamshire, UK, RPN2106). Densitometric scanning analysis was conducted by NIH Image J 1.62 software on Mac OS X version 10.15.7 (Apple Computer International, Cupertino, CA, USA).

### 4.3. Extracellular Vesicle Isolation from Cultured Cells

EVs were prepared by differential centrifugation [34]. Briefly, cells were first incubated in the EV-depleted medium obtained after the centrifugation of RPMI at 100,000× *g* at 4 °C overnight and filtering of the supernatant at 0.2 μm to remove any particles from FBS. Then, after 24 h, conditioned medium was collected from approximately 95% confluent in cell culture flasks, either untreated or subjected to a 4 h treatment with HBSS at 37 °C. The collected media underwent an initial centrifugation at 400× *g* for 10 min at room temperature (RT) to precipitate and eliminate cells. All the following centrifugation steps were carried out at 4 °C. Subsequently, the supernatant was spun at 2000× *g* for 20 min to remove debris and apoptotic bodies. Following that, for the collection of large EVs (lEVs), the supernatant was centrifuged at 15,000× *g* for 40 min. The resulting lEV pellet (P15) was reconstituted in a large volume of phosphate-buffered saline (PBS) and subjected to ultracentrifugation at 15,000× *g* for 40 min to wash the sample. To eliminate any residual lEVs, the media supernatant from the first 15,000× *g* step was filtered through a 0.22 μm pore PES filter (Millipore, Burlington, MA, USA). This supernatant (pre-cleared medium) was next subjected to ultracentrifugation at 120,000× *g* for 4 h in a SW 32 Ti Rotor Swinging Bucket rotor (k factor of 204, Beckman Coulter, Fullerton, CA, USA) to sediment small EVs (sEVs). The crude sEV pellet (P120) was resuspended in a large volume of PBS, followed by ultracentrifugation at 120,000× *g* for 4 h to purify the sample.

### 4.4. Concentration and 6–30% Iodixanol Density Gradient Fractionation Analysis of sEVs

The concentration of sEVs (P120) was measured by TEM from untreated and HBSS-treated cells. For each group, 12 independent images of randomly chosen fields at the same magnification (60,000×) were analyzed and vesicles counted in a selected area of 1 μm^2^. The total number of vesicles was normalized for the volume used for the grid preparation (3 μL). This approach is useful for a comparison based solely on the resolution of the microscopy method used.

The crude pellets of sEVs (P120) were then subjected to 6–30% iodixanol density gradient fractionation. Samples were reconstituted in ice-cold PBS and combined with an ice-cold iodixanol/PBS solution, resulting in a final 30% iodixanol concentration. This suspension was then carefully added to the bottom of a centrifugation tube. To create the complete gradient, solutions containing decreasing concentrations of iodixanol in PBS were layered on top. Ultracentrifugation was performed at 120,000× *g* for 15 h at 4 °C using a SW41 rotor (Beckman Coulter). From the top of the gradient, 12 individual 1 mL fractions were collected.

### 4.5. Transmission Electron Microscopy and Immunogold Analysis

On ice, cells were fixed in 2.5% glutaraldehyde in 0.1 mol/L cacodylate buffer pH = 7.4 for 1 h and post-fixed in 1% OsO_4_ in the same buffer for 2 h. After fixation, cells were dehydrated with ethanol (25%, 50%, 70%, 90%, and 100%) and embedded in Spurr resin. Then, 60 nm sections were examined under the electron microscope (Zeiss, Oberkochen, Germany).

For EV imaging, all fractions were fixed with 0.1% paraformaldehyde in PBS (30 min, RT). Fixed EVs were loaded on formvar-carbon-coated grids, stained with 2% uranyl acetate (7 min, RT), and observed. For immunogold labeling, fixed EVs were loaded on formvar-carbon grids by inversion and then floated sequentially on droplets of PBS (three times for 3 min), on droplets of PBS with 0.5% bovine serum albumin (BSA) (10 min), primary antibody (60 min), 1% BSA (six times for 3 min), secondary antibody-or protein A-gold conjugate (40 min) and 1% BSA (three times for 3 min), and on droplets of PBS (three times for 3 min). Primary antibodies were used at the following concentrations: 10 μg/mL of anti-LC3B-II (Novus Biologicals, NB600-1384SS) and 10 μg/mL of anti-ganglioside GD3 (Abcam, Cambridge, UK, ab11779). Colloidal gold-conjugated goat anti-rabbit IgG H&L (Abcam, 6 nm, ab41498) and protein A Gold Conjugate (Abcam, 20 nm, ab270601) were employed. Then, after the washing steps, samples were contrasted with 2% uranyl acetate (7 min, RT) and examined. For all the observations, the electron microscope Zeiss Auriga Scanning Electron Microscope (Zeiss, Oberkochen, Germany) equipped with a STEM module and operating at 20 kV was used.

### 4.6. Atomic Force Microscopy Preparation Procedure, Imaging, and Analysis

For atomic force microscopy, untreated and HBSS-treated cells for 4 h at 37 °C were fixed in 1% glutaraldehyde in 0.1 mol/mL cacodylate buffer pH 7.4 at ice temperature for 30 min. After fixation, cells were washed again twice in DPBS, and two final washes in bidistilled water were performed. The AFM images were acquired in contact mode with a Multimode AFM (Bruker, Santa Barbara, CA, USA). We used commercially available AFM tips, model DNP (Bruker), with a triangular-shaped cantilever, a nominal elastic constant of 0.06 N/m, and a nominal tip radius of 10 nm. The interaction force between the tip and the sample was selected to be less than one nN, in such a way as to not damage both tip and sample. To gain an overview of the sample morphology, two different strategies were adopted: when imaging cells, the scan area was set to 60 μm, and a subsequent zoom of 20 μm was performed on an area of interest. Representative 3D reconstructions and mapping of the cell areas (6 × 6 μm^2^) of AFM images were performed with the Gwyddion software version 2.66. All images were minimally treated (i.e., a mean plane subtraction, substrate tilt correction, and line coupling) to avoid thermal drift and the tilting of the sample. For the mapped regions (nuclear and perinuclear to cellular edges), squared 6 × 6 μm^2^ areas were randomly selected, and five measurements were taken from each sample. The roughness value is a quantitative parameter and thus can be used to compare different samples and regions if it is calculated over areas with the same lateral dimension and the same total number of pixels. For roughness measurements, AFM surface root mean square deviation (roughness; Rq) analysis of the cell surface was performed by ImageJ Software version 1.8.0.

### 4.7. Nanoparticle Tracking Analysis (NTA)

The concentration and the size distribution of sEVs from untreated cells or from HBSS-treated cells in the presence or absence of bafilomycin A1 (Baf A1; 1 μM) were measured by NTA analysis. EV pellets were resuspended in PBS, and 20 µL were used for NTA analysis by a NS300 instrument (Malvern, UK, Software Version 3.4). To get a suitable concentration, the samples were diluted in freshly filtered PBS 2.5 × 10^2^ and vortexed for 1 min. Samples were flowed at a syringe pump speed of 30 AU.

### 4.8. Western Blot Analysis of 6–30% Iodixanol-Gradient Fractions

The collected fractions were subjected to SDS-PAGE. Of note, for CD63 and CD81 detection, fractions were subjected to SDS-PAGE under nonreducing conditions [34]. The proteins were then transferred onto PVDF membranes (Bio-Rad, 162–0177). To prevent nonspecific binding, the membranes were blocked with a solution containing 1% BSA in TBS (Bio-Rad, 1706435) with 0.05% Tween 20 (Bio-Rad, 1706531). The membranes were probed with rabbit anti-CD63 antibodies (Abcam, ab216130), rabbit anti-CD81 (Abcam, ab155760), anti-Annexin A1 (Santa Cruz Biotechnology, sc-12740) antibodies, rabbit anti-ALIX (Abcam, AB22555), anti-GM130 (Santa Cruz, SC55591), rabbit anti-ERLIN1 antibodies (Thermo Fisher Scientific, Waltham, MA, USA, PA5-17102), mouse anti-LC3-II antibodies (Abcam, ab243506), or mouse anti-GD3 R24 antibodies (Abcam, ab11779). The interaction was visualized using horseradish peroxidase (HRP)-conjugated secondary antibodies. Immunoreactivity was detected through a chemiluminescence reaction using the ECL western detection system (Amersham, RPN2106). Densitometric scanning analysis was performed using NIH Image J 1.62 software on Mac OS X (Apple Computer International, Cupertino, CA, USA).

### 4.9. Immunoprecipitation Experiments

Exosomes obtained from both untreated or HBSS-treated 2F cells were lysed in lysis buffer (10 nM TRIS-HCl, pH 8.0, 150 mM NaCl, 1% Nonidet P-40, 1 mM phenylmethylsulfonyl fluoride (PMSF), and 10 mg/mL leupeptin). The lysates were mixed with protein A/G-acrylic beads (Sigma-Aldrich, P3296) and stirred by a rotary shaker for 2 h at 4 °C to preclear nonspecific binding. After centrifugation (500× *g* for 1 min), the supernatant was recovered and immunoprecipitated with mouse anti-LC3-II antibodies plus protein A/G-acrylic beads. A mouse IgG isotypic control (Sigma, I5006) was used. The immunoprecipitates were checked by dot blot and Western blot analysis.

### 4.10. GD3 Immunostaining in the LC3-II Immunoprecipitated from Exosomes

In brief, aliquots obtained from immunoprecipitates of LC3-II of treated or untreated 2F cells with HBSS were spotted onto nitrocellulose strips and subjected to dot blot analysis, which represents the most suitable methodological approach to detecting ganglioside molecules [28]. The nitrocellulose strips were subsequently incubated with IgG anti-GD3 R24 antibodies (Abcam, ab11779) for 1 h at room temperature. Following incubation, the strips were washed in PBS and subsequently incubated for 1 h at 37 °C with HRP-conjugated anti-mouse IgG. Immunoreactivity was then evaluated using the ECL western detection system (Amersham, RPN2106) to detect chemiluminescence.

### 4.11. Statistical Analysis

The statistical analysis was performed using GraphPad Prism software Inc. version 9.5.0 (San Diego, CA, USA). All reported data in this paper underwent rigorous verification through three independent experiments performed in duplicate, and the results are presented as mean ± standard deviation (SD). The *p*-values for all graphs were determined using the Student’s *t*-test, as indicated in the figure legends. Significance levels were denoted by asterisks: * *p* < 0.05, ** *p* < 0.005, *** *p* < 0.001, and **** *p* < 0.0001, corresponding to increasing levels of significance.

## 5. Conclusions

In conclusion, this study, by using different methodologies for EV morphological analysis, deals with the generation of intracellular and extracellular vesicles triggered by the autophagic process. A relevant finding is the presence of lipid raft components within secreted EVs and their physical association with the LC3-II marker. It might prompt the development of alternative raft target therapies for diseases in which the generation of EVs is active.

## Figures and Tables

**Figure 1 ijms-25-06175-f001:**
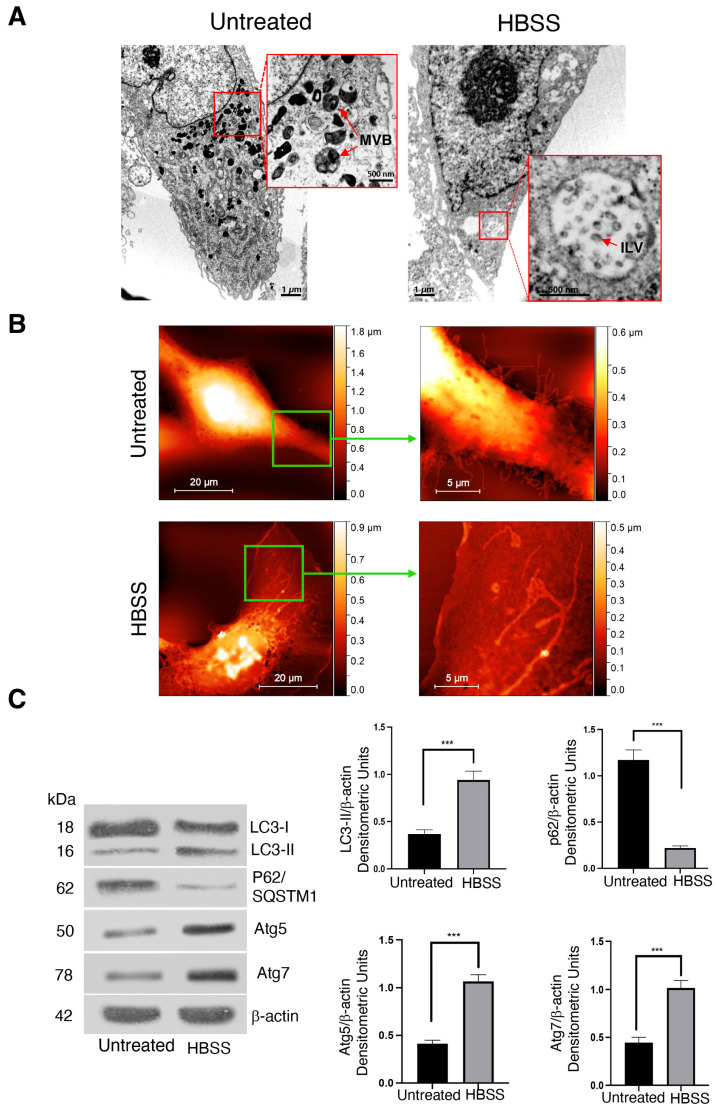
Autophagy triggers the formation of intracellular enlarged structures in HT1080-derived fibrosarcoma 2FTGH cells. (**A**) HT1080-derived 2FTGH cells, untreated or treated with HBSS for 4 h at 37 °C, were analyzed using TEM. In untreated cells, heterogeneous MVB-like structures were detectable in proximity to the plasma membrane (scale bar: 1 μm) (red arrows in red box, 500 nm); in treated cells large MVB-containing ILVs are visible in the cytoplasm (scale bar: 1 μm) (red arrow in red box, scale bar: 500 nm). MVB = multivesicular bodies; ILV = intraluminal vesicle. (**B**) AFM images of control and HBSS-treated cells (scale bar: 20 μm). Green boxes represent the magnification area of the image (scale bar: 5 μm). (**C**) Cells, untreated or treated with HBSS for 4 h at 37 °C, were lysed in lysis buffer. After evaluation of the protein concentration, the lysate was subsequently analyzed by immunoblotting using rabbit anti-LC3, rabbit anti-p62/SQSTM1, rabbit anti-Atg5, rabbit anti-Atg7, or mouse anti-β-actin antibodies. A representative experiment among 3 is shown. The bar graph shows densitometric analysis. Results represent the mean ± SD. *** *p* < 0.001.

**Figure 2 ijms-25-06175-f002:**
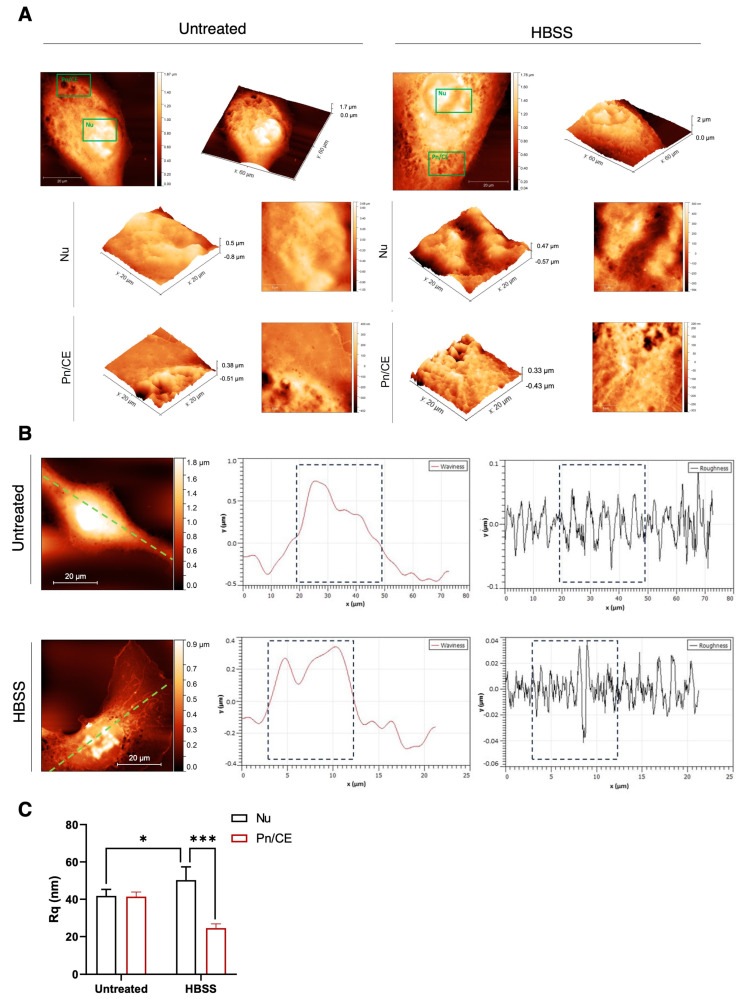
AFM topography, 3D reconstructions, and surface roughness. (**A**) Typical AFM images acquired on control and HBSS-treated cells in 2D and 3D. For each image (scale bar: 20 μm), a zoom on a region of interest highlighted in green is reported in 2D (right) (scale bar: 6 μm) and 3D (left and below) views in order to better visualize the spatial arrangement of the membrane. (**B**) Example of roughness and waviness analysis. (left side) AMF images (scale bars 5 μm) of untreated (top line) and HBSS (bottom line). The profiles were performed along the green dash line drawn on the images and are reported on the central side (waviness) and right side (roughness). The waviness allows to discriminate between the sample regions (Nu or Pn/CE), while the roughness allows to estimate the membrane status. The nuclear regions were highlighted in the dashed boxes; the roughness depends only on the membrane arrangement and not from the cellular morphology beneath. (**C**) Roughness values obtained from all the images sampled of a region (Nu or Pn/CE) for each cell. The roughness values were statistically different between nuclear and perinuclear regions (*** *p* < 0.001) in the HBSS-treated samples and in the nuclear region between control and HBSS-treated cells (* *p* < 0.05).

**Figure 3 ijms-25-06175-f003:**
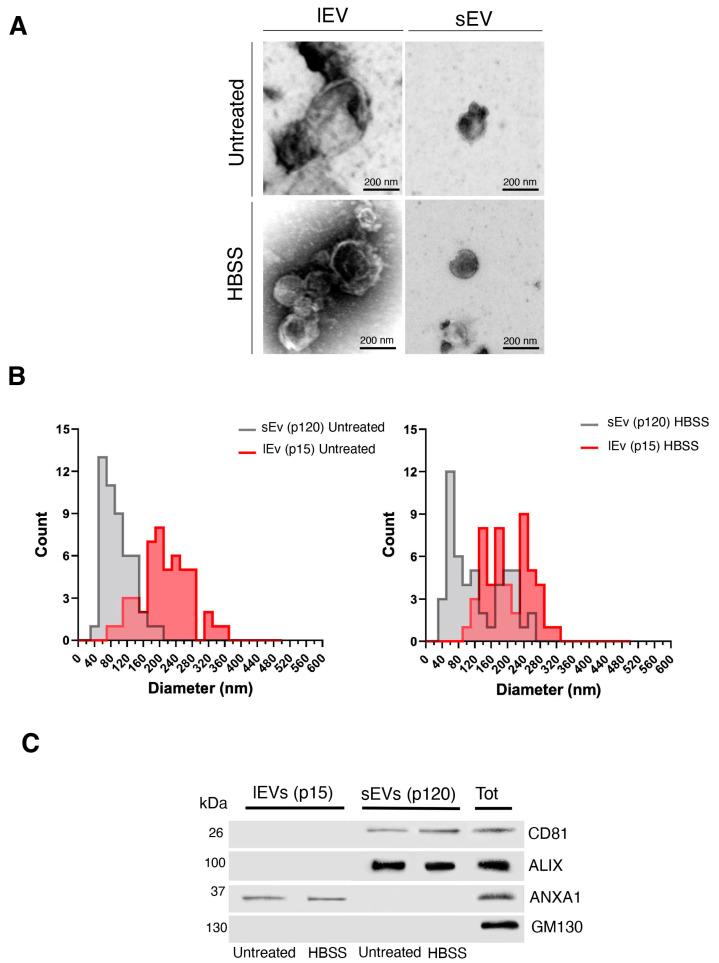
Characterization of small and large EVs released by HT1080-derived 2FTGH cells after autophagy stimulation. (**A**) Representative TEM images of large EVs (p15, lEVs) and small EVs (p120, sEVs) fractions obtained from 2FTGH cells, untreated or treated with HBSS for 4 h at 37 °C (scale bar: 200 nm). (**B**) The size distribution of large EVs (p15, lEVs) and small EVs (p120, sEVs) from HBSS-treated (4 h at 37 °C) or untreated cells was obtained by measuring at least 60 vesicles for each sample in 5 randomly selected microscopic fields. (**C**) Immunoblots of large EVs (lEVs) and small EVs (sEVs) fractions from cells, untreated or treated with HBSS, using anti-CD81, anti-ALIX, anti-Annexin A1 (ANXA1), and anti-GM130 antibodies. A representative experiment among 3 is shown.

**Figure 4 ijms-25-06175-f004:**
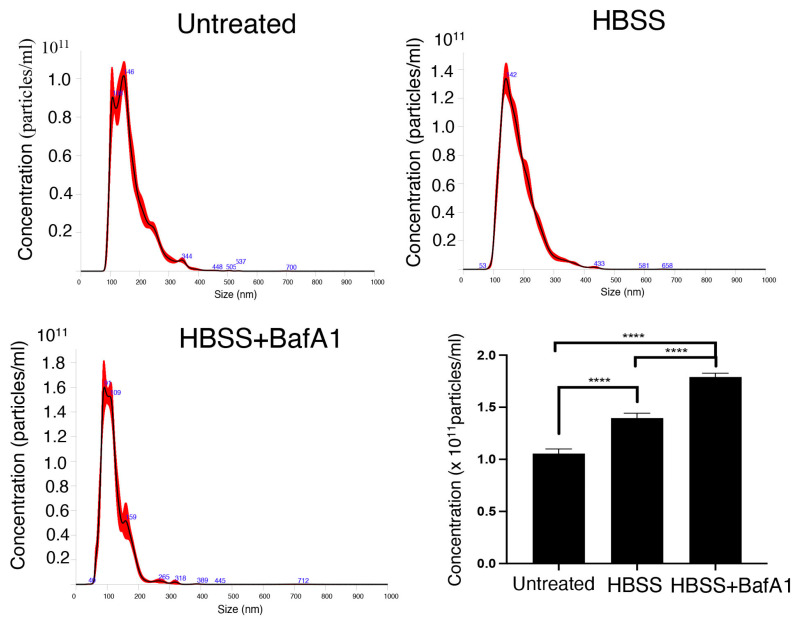
Concentration of small EVs measured by NTA analysis. NTA quantification of small EVs, expressed as particles/mL released from untreated, HBSS-treated, or HBSS-BafA1-treated cells. Values are means ± SD (*n* = 5). **** *p* < 0.0001.

**Figure 5 ijms-25-06175-f005:**
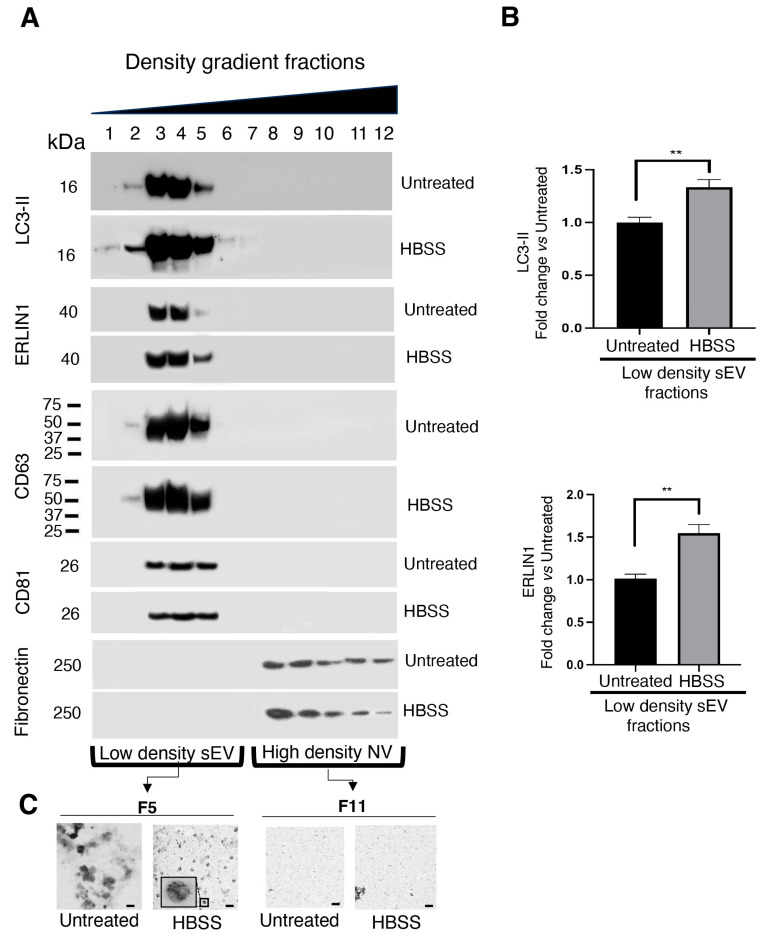
Analysis of small EVs by 6–30% iodixanol gradient density. (**A**) Immunoblot analysis of iodixanol density gradient-purified fractions obtained from HT1080-derived 2FTGH cells, untreated or treated with HBSS for 4 h at 37 °C. Twelve fractions, corresponding to low-density fractions (3–5) indicative of small EVs and high-density fractions (8–12) enriched in non-vesicular (NV) components, were probed with mouse anti-LC3-II antibodies, with rabbit anti-ERLIN1 antibodies, with rabbit anti-CD63 antibodies, with rabbit anti-CD81 antibodies, or with rabbit anti-fibronectin. The observed MW may vary in the range of 25–65 (predicted MW: 26 kDa), depending on post-translational modifications, post-translational cleavages, relative charges, and other experimental factors, including the unreducing condition. A representative experiment among 3 is shown. (**B**) The bar graph shows densitometric analysis. Results represent the mean ± SD. ** *p* < 0.005. (**C**) Representative TEM images of iodixanol density gradient-purified fractions 5 (F5) and 11 (F11) obtained from 2FTGH cells, untreated or treated with HBSS for 4 h at 37 °C (scale bar: 200 nm).

**Figure 6 ijms-25-06175-f006:**
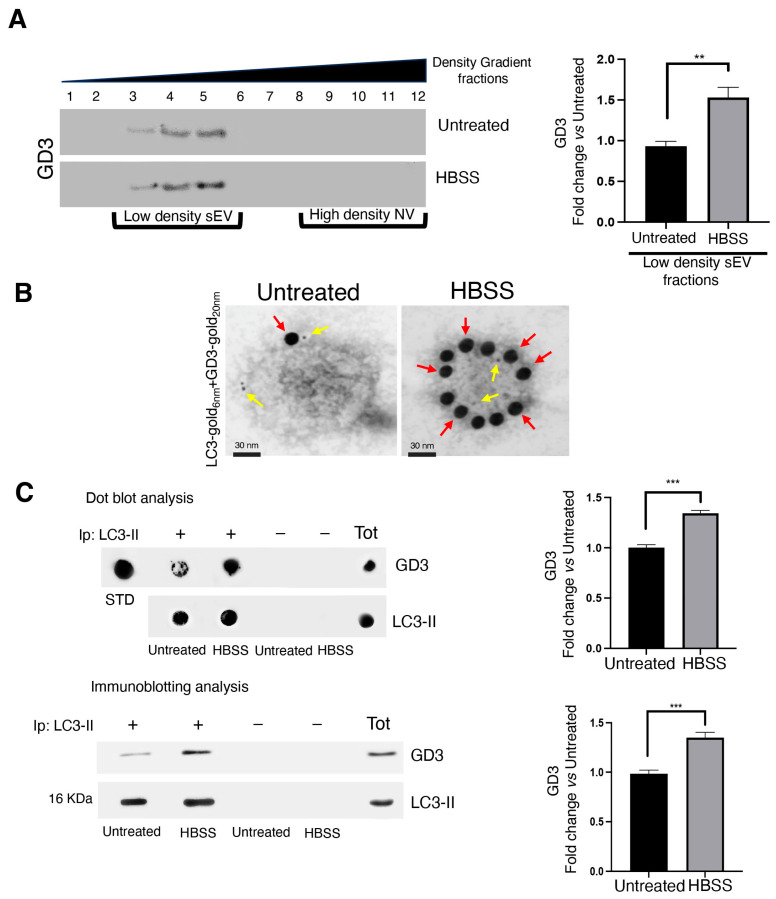
Stimulation of autophagy leads to the association of LC3-II and GD3 within small EVs. (**A**) Immunoblot analysis of iodixanol density gradient-purified fractions obtained from cells, untreated or treated with HBSS for 4 h at 37 °C. Twelve fractions were probed with anti-GD3 R24 antibodies, as described in the Materials and Methods section. The loading control is shown in Figure 5A. The bar graph shows densitometric analysis. Results represent the mean ± SD of 3 independent experiments. ** *p* < 0.005. (**B**) Representative TEM images of the immunogold double labeling of sEVs from 2FTGH cells, untreated (scale bar: 30 nm) or treated with HBSS for 4 h at 37 °C (scale bar: 30 nm) to detect LC3-II (6 nm gold particles, yellow arrows) and GD3 (20 nm gold particles, red arrows) at the EV surface. (**C**) Exosomes obtained from both untreated or HBSS-treated 2FTGH cells were lysed in lysis buffer (10 nM TRIS-HCl, pH 8.0, 150 mM NaCl, 1% Nonidet P-40, 1 mM phenylmethylsulfonyl fluoride, and 10 mg/mL leupeptin). The lysates were immunoprecipitated with mouse anti-LC3-II (+) or with IgG control (−). The immunoprecipitates were analyzed by dot blot and Western blot analysis using anti-GD3. STD = pure standard GD3. Bar graphs show densitometric analyses. Results represent the mean ± SD of 3 independent experiments. *** *p* < 0.001.

**Table 1 ijms-25-06175-t001:** Concentration of small EVs measured by TEM from untreated and HBSS-treated cells. For each group, 12 independent images of randomly chosen fields at the same magnification (60,000×) were analyzed and vesicles counted in a selected area of 1 μm^2^. The total number of vesicles was normalized for the volume used for the grid preparation (3 μL). * *p* < 0.05.

	Small EVs/μm^2^/μL	Total Small EVs/μL
Untreated	7.02 ± 1.9	84
HBSS	9.4 ± 2.8 *	113

## Data Availability

The datasets generated and/or analyzed during the current study are not publicly available but are available from the corresponding author on reasonable request.

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
