# Peer review of "Autophagy Promotes Enrichment of Raft Components within Extracellular Vesicles Secreted by Human 2FTGH Cells"

_ijms, 2024, doi:10.3390/ijms25116175_

Round 1
Reviewer 1 Report (Previous Reviewer 3)
Comments and Suggestions for Authors
Second´s comments
Original article: Autophagy Promotes Enrichment of Raft Components within Extracellular Vesicles Secreted by Human 2FTGH Cells
Authors: Valeria Manganelli, Luciana Dini, Stefano Tacconi, Simone Dinarelli, Antonella Capozzi, Gloria Riitano, Serena Recalchi, Tuba Rana Caglar, Roberta Misasi, Maurizio Sorice and Tina Garofalo
I appreciate the effort of the authors to improve the manuscript.
There are still several points to be made in full or to answer and justify ones of the observations made.
To my comments -The in vitro model should justify-
I consider it important to justify in the introduction with more than one reference (#17)...The objective of the study. The introduction does not specifically address sarcoma. The examples cited in general are in non-carcinogenic processes, so, are the mechanisms of autophagy and its relationship with Raft Components within Extracellular Vesicles Secreted similar in normal cells? Or are the mechanisms of autophagy similar to those of carcinogenic cells as in the case of human sarcoma cell line HT 1080? Therefore, the article can be further strengthened and clearly stated if there are adjustments in the introduction.
In response to reference 24, Regulation of ULK1 Expression and Autophagy by STAT1. Actually, human sarcoma cell line HT 1080, is not taken as a control and considered as a non-transfected cell, but rather as a control in case of comparison of transformed lines, STAT1-deficient human fibrosarcoma (U3A) cells, or STAT1-deficient human fibrosarcoma cells reconstituted with STAT1 (U3A-R). Therefore, it is still recommended to perform all assays with normal fibroblasts.
In response to reference 25. the human sarcoma cell line HT 1080 was used as a control but in the case of Type 2 transglutaminase (TG2) knockout mice display, which like reference 24 applies in the case of transformed cells or animal models. Therefore, there is still a doubt as to whether the modifications presented by the cell line used do not affect the autophagy process and in fact reflect a process similar to autophagy in normal cells or whether it is similar and representative of a process linked to the human sarcoma cell line HT 1080.
To my comment: “In the introduction, everything concerning autophagy and the formation of extracellular vesicles as key molecules is presented, so it is suggested to change the cell line of study to a non-transformed cell line or primary culture”.
Please present evidence in introduction section that support your objective of to analyze the role of autophagy on raft components within EVs only in a tumorigenic cell line since in tumor cells EVs are known to have a significant impact on migration, invasion and metastasis and associated with your aim that was to analyze whether rafts components are enriched within EVs cargoes
To my comment: Figure 1A for the untreated group only one region shows protrusions and buds in the first image and for the rest of the micrographs of that group no protrusions and buds are observed.
Your answer: We agree with the reviewer that Figure 1A does not prompt to state that several protrusions and buds on the cell surface were observed. We modified the text accordingly.
The text has not been modified in its entirety and is still as follows
"conversely, note the absence of protrusions in HBSS-treated cells (arrows)".
In reality, in Figure 1A concerning the micrographs of 2FTGH untreated cells, only 1/3 present protrusions.Therefore, it is again questioned whether this observation is important and whether it occurs in all 2FTGH untreated cells.
To my comment: “When comparing treatments or samples the micrographs should be of the same size, at the same magnification with the same scale bar. Images with inconsistencies in the scale bar or in the size of the image presented are shown below”.
The images are not presented with uniform enlargements.
There is still room for improvement, in Figure 1 A, the ILV have different scale and the value of the scale bars is not indicated in any part of the paper.
It is important to remember that if you present comparison results between untreated and HBSS treated you should present images taken at the same scale so it is expected to visualize at least the ILV micrographs under the same capture conditions.
Figure 3A, bar scale.
Figure 5B, bar scale is different, still showing HBSS treated at a higher magnification. It is necessary to compare the images at the same resolution and size.
To my comment: Figures 3-C and 4-A report different isoforms for CD63, what is the biological explanation for reporting different isoforms.
The authors should answer what is the biological explanation for reporting different isoforms. The authors should show both isofomars for Figures 3-C and 4-A, or only one isoform for the different images in order to have homogeneous and demonstrative results.
To my comment: All images of spotted onto nitrocellulose strips should be performed in an electrophoretic run and send complete evidence of the membranes revealed in WB.
The dot blot methodology has the disadvantage of lack of precise quantification and the possibility of false positive or false negative results. In your answer: "In these experiments non proteic molecules were detected", the results of non-detection are the result of low sensitivity or specific results. Dot blots are not designed to quantify, dot blots have an absence of a loading control protein, in dot blots there is no way to demonstrate that in all cases the same amount of protein was deposited. It is doubtful whether the result is a false positive. It is suggested to obtain WB images for GD3 or to perform an additional technique, such as immunofluorescence technique. Also Figure 5B can be enriched with more micrographs of at least 3 for each of the experimental conditions.
I thank again the authors for improving the manuscript, I hope they will improve the points that were left unchanged in order to have a quality version and consider its publication.
Author Response
To my comments -The in vitro model should justify-
In a previous study we analyzed the role of raft components in the regulation of autophagic process using HT1080-derived human fibrosarcoma 2FTGH (2F) cells (Manganelli et al., Autophagy 2021). Thus, in the present study, we used the same cell line to investigate the enrichment of raft components within EVs induced by autophagy. Of note, this cell line is usually employed in several studies (Goldberg et al, J Biol Chem 2017; Manganelli et al., Autophagy 2021; Manganelli et al., Biomolecules 2021) and the autophagy process was analyzed in detail in these cells.
1 I consider it important to justify in the introduction with more than one reference (#17)...The objective of the study. The introduction does not specifically address sarcoma. The examples cited in general are in non-carcinogenic processes, so, are the mechanisms of autophagy and its relationship with Raft Components within Extracellular Vesicles Secreted similar in normal cells? Or are the mechanisms of autophagy similar to those of carcinogenic cells as in the case of human sarcoma cell line HT 1080? Therefore, the article can be further strengthened and clearly stated if there are adjustments in the introduction.
We thank the reviewer for these interesting suggestions. We improved the Introduction section, adding new references (n.17,18,19), which describe the release of EVs by cancer cells and clarify the role of some raft components within EVs released in pathological processes. Interestingly, these studies show a high enrichment of lipid species from cells to exosomes mainly in glycosphingolipids, sphingomyelin, cholesterol, and phosphatidylserine.
2 In response to reference 24, Regulation of ULK1 Expression and Autophagy by STAT1. Actually, human sarcoma cell line HT 1080, is not taken as a control and considered as a non-transfected cell, but rather as a control in case of comparison of transformed lines, STAT1-deficient human fibrosarcoma (U3A) cells, or STAT1-deficient human fibrosarcoma cells reconstituted with STAT1 (U3A-R). Therefore, it is still recommended to perform all assays with normal fibroblasts.
As reported above, we validated the use of this cell line as a suitable in vitro model for analyzing the role of raft components during the autophagic process in two previous publications (Manganelli et al., Autophagy 2021; Manganelli et al., Biomolecules 2021), demonstrating that autophagic machinery, including ULK1 activity, was not different from normal fibroblasts.
3 In response to reference 25. the human sarcoma cell line HT 1080 was used as a control but in the case of Type 2 transglutaminase (TG2) knockout mice display, which like reference 24 applies in the case of transformed cells or animal models. Therefore, there is still a doubt as to whether the modifications presented by the cell line used do not affect the autophagy process and in fact reflect a process similar to autophagy in normal cells or whether it is similar and representative of a process linked to the human sarcoma cell line HT 1080.
In previous studies (Goldberg et al, J Biol Chem 2017; Manganelli et al., Autophagy 2021; Manganelli et al., Biomolecules 2021), the autophagy process was analyzed in detail in these cells, demonstrating that it is not affected.
4 To my comment: “In the introduction, everything concerning autophagy and the formation of extracellular vesicles as key molecules is presented, so it is suggested to change the cell line of study to a non-transformed cell line or primary culture”.
Please present evidence in introduction section that support your objective of to analyze the role of autophagy on raft components within EVs only in a tumorigenic cell line since in tumor cells EVs are known to have a significant impact on migration, invasion and metastasis and associated with your aim that was to analyze whether rafts components are enriched within EVs cargoes
We improved the Introduction section according to the reviewer’s suggestions, supporting the focus of this investigation on the role of autophagy on raft components within EVs only in a tumorigenic cell line, indicating that : a) cancer cells release more exosomes which contribute to many aspects of tumors malignity, including proliferation, invasion, metastasis, angiogenesis and immunosuppression; b) a high enrichment of lipid species from cancer cells to exosomes.
To my comment: Figure 1A for the untreated group only one region shows protrusions and buds in the first image and for the rest of the micrographs of that group no protrusions and buds are observed.
Your answer: We agree with the reviewer that Figure 1A does not prompt to state that several protrusions and buds on the cell surface were observed. We modified the text accordingly.
The text has not been modified in its entirety and is still as follows
"conversely, note the absence of protrusions in HBSS-treated cells (arrows)".
In reality, in Figure 1A concerning the micrographs of 2FTGH untreated cells, only 1/3 present protrusions. Therefore, it is again questioned whether this observation is important and whether it occurs in all 2FTGH untreated cells.
TEM and AFM are two different modalities for obtaining morphological images of cells. Precisely because of the way the cells are prepared and the fact that they are 2D thin sections where the main purpose is to show the MVBs (internal structures visible only after TEM sectioning), it is not possible to calculate from the photos that only one third of the cell has protrusions. This information can only be obtained by long tomographic reconstructions. This is precisely why AFM analysis was carried out, which allows the entire surface of the cells to be visualised and then the distribution of budding, protrusions, etc. of the cells to be calculated by topographic mapping. In this way, it was possible to show a different presence of surface budding between control cells and after induction of autophagy. In addition, the AFM data reported in the paper also showed a different distribution of budding between peripheral and nuclear parts of the cell. Finally, AFM topography allows us to analyse a large number of cells and thus obtain very reliable data for the best statistics.
Anyway, we agree with the reviewer’s comment and decided to replace Figure 1A with the aim to better clarify the message. In particular, in the new Fig. 1A we show only one representative TEM micrograph of untreated cells and one of HBSS-treated cells. The analysis performed by AFM is showed in Fig. 1B.
To my comment: “When comparing treatments or samples the micrographs should be of the same size, at the same magnification with the same scale bar. Images with inconsistencies in the scale bar or in the size of the image presented are shown below”.
The images are not presented with uniform enlargements.
There is still room for improvement, in Figure 1 A, the ILV have different scale and the value of the scale bars is not indicated in any part of the paper.
It is important to remember that if you present comparison results between untreated and HBSS treated you should present images taken at the same scale so it is expected to visualize at least the ILV micrographs under the same capture conditions.
We have adjusted the scales in the Figure. The MVB appears to be the same size in both untreated and treated cells.
Figure 3A, bar scale.
We have added bar scale in Figure 3A and not only in the legend.
Figure 5B, bar scale is different, still showing HBSS treated at a higher magnification. It is necessary to compare the images at the same resolution and size.
We have adjusted the scale in the new Figure 6B.
To my comment: Figures 3-C and 4-A report different isoforms for CD63, what is the biological explanation for reporting different isoforms.
The authors should answer what is the biological explanation for reporting different isoforms. The authors should show both isoforms for Figures 3-C and 4-A, or only one isoform for the different images in order to have homogeneous and demonstrative results.
The right molecular weight of CD63 detected by anti-CD63 antibodies (Abcam, ab216130) is 26 kDa, as reported in the old Fig 4A. Sorry, we now reported the right MW in the new Fig. 3C.
To my comment: All images of spotted onto nitrocellulose strips should be performed in an electrophoretic run and send complete evidence of the membranes revealed in WB.
The dot blot methodology has the disadvantage of lack of precise quantification and the possibility of false positive or false negative results. In your answer: "In these experiments non proteic molecules were detected", the results of non-detection are the result of low sensitivity or specific results. Dot blots are not designed to quantify, dot blots have an absence of a loading control protein, in dot blots there is no way to demonstrate that in all cases the same amount of protein was deposited. It is doubtful whether the result is a false positive. It is suggested to obtain WB images for GD3 or to perform an additional technique, such as immunofluorescence technique. Also Figure 5B can be enriched with more micrographs of at least 3 for each of the experimental conditions.
We added in the new Figures 5A and 5C Western blot and densitometric analysis. We added a new micrograph in Supplementary Figure 1.
Reviewer 2 Report (Previous Reviewer 1)
Comments and Suggestions for Authors
The authors need to perform some experiments (Nanoparticle tracking analysis). Likewise, in the images in Figure 1A, in both the experimental and the control, no differences are observed; In both conditions, multivesicular bodies are observed close to the plasma membrane.
Author Response
The authors need to perform some experiments (Nanoparticle tracking analysis).
We thank the reviewer for the reminder. The required analysis by NTA has been carried out. The data are reported in the text (Results section, new Figure 4 and Materials and Methods section).
Likewise, in the images in Figure 1A, in both the experimental and the control, no differences are observed; In both conditions, multivesicular bodies are observed close to the plasma membrane.
The proximity of the MVBs to the plasma membrane is due to the fact that the sections observed are longitudinal sections of fibroblasts adhering to the surface of the culture plate. Due to their nature, fibroblasts have an extremely conical and flattened shape. This means that the cytoplasm has a reduced thickness, which results in MVBs being close to the plasma membrane, regardless of whether ctrl or treated cell is observed.
Anyway, we decided to replace Figure 1A with the aim to better clarify the message. In particular, in the new Fig. 1A we show only one representative TEM micrograph of untreated cells and one of HBSS-treated cells.
Reviewer 3 Report (New Reviewer)
Comments and Suggestions for Authors
The current study entitled “Autophagy Promotes Enrichment of Raft Components within Extracellular Vesicles Secreted by Human 2FTGH Cells” by Valeria Manganelli et al is aimed to study crosstalk between autophagy and endo-lysosomal system. In the current article they have explore the autophagy induced extracellular vesicles (EVs) in human fibrosarcoma 2FTGH cells, using Transmission Electron Microscopy (TEM) and Atomic Force Microscopy (AFM) with unconventional secretion pathway in the process of autophagy. This study is extension of the previously published (doi: 10.1080/15548627.2020.1834207) from the same group, where they had reported that AMBRA1-ERLIN1 interaction within MAM raft-like microdomains appears to be pivotal in promoting the formation of autophagosomes. However, there are several points that need to be addressed are listed below and hope those comments would be helpful to improve the quality of article in future:
Major Points:
1. Although these studies are interesting, however, I would like to criticize authors for using only one cell line Human 2FTGH (2F) fibroblasts to explore this important topic (any specific reason).
2. Execute experiment to analyze if removal of important component from lipid raft (GD3) may alter autophagic function (activation or reversal?)
3. Lysosomal pH alteration during EV genesis and autophagic process (perform experiment with inhibitor for downstream signaling)
Minor Points:
Figure 1:
A- In TEM images, authors must illustrate other important cellular component like mitochondria, ER, Golgi and nucleus to have better understanding of the structural composition of cells.
B- Western blot expression of ATG5 and ATG7 should be perform in the same conditions.
Figure 5: C-Control IgG is missing
Author Response
Major Points:
- Although these studies are interesting, however, I would like to criticize authors for using only one cell line Human 2FTGH (2F) fibroblasts to explore this important topic (any specific reason).
In previous studies we validated the use of this cell line as a suitable in vitro model for analyzing the role of raft components during the autophagic process (Manganelli et al., Autophagy 2021; Manganelli et al., Biomolecules 2021). The current work represents, in a certain sense, an in-depth study in the same direction, reserving the amplification of the applicative and translational systems for subsequent work, which, as suggested, is of particular interest in the neoplastic field.
- Execute experiment to analyze if removal of important component from lipid raft (GD3) may alter autophagic function (activation or reversal?)
The reviewer's observation is absolutely relevant and in line with our approach to this type of experimentation. In fact, in one of our previous papers (Autophagy 2016, doi: 10.1080/15548627.2020.1834207) the evidence of how HBSS-induced autophagy was significantly reduced in GD3 synthase ST8SIA1 silenced cells, as compared to scrambled siRNA-transfected cells was clearly stated. Moreover, a direct role of GD3 in the biogenesis and maturation of autophagic vacuoles under nutrient deficiency had already been reported few years earlier [Autophagy 2014 doi: 10.4161/auto.27959]. Indeed, it was revealed that GD3 can be detected in immature autophagosomes, associated with phosphatidylinositol 3-phosphate (PI3P) and LC3-II, as well as in autolysosomes, associated with LAMP1. Blocking sphingolipids biogenesis after knockdown of GD3 synthase affected the formation of autophagosomes, hindering autophagic flux. We believe that the data acquired is solid. Further studies are in progress to analyze the effective role of lipid raft components (i.e. GD3) on EVs release.
- Lysosomal pH alteration during EV genesis and autophagic process (perform experiment with inhibitor for downstream signaling)
We analyzed the effect of Bafilomycin A1 on EVs release by NTA analysis. In particular, to monitor whether functional inhibition of autophagosome/lysosome fusion alters the rate of EVs release, we tested the effects of HBSS in combination with the lysosomal inhibitor Baf A1. We found that Baf A1 treatment increased EVs release from 2FTGH human fibrosarcoma cells. We added this finding in the new Figure 4, in the Results and Discussion sections.
Minor Points:
Figure 1:
A- In TEM images, authors must illustrate other important cellular component like mitochondria, ER, Golgi and nucleus to have better understanding of the structural composition of cells.
Illustrating the other cellular components, such as mitochondria, ER, Golgi and nucleus, is not conducive to a better understanding of the structural composition of the cell. In fact, we believe that depicting all the structures of a cell by filling in symbols and letters prevents one from grasping what the organelles under investigation are. MVBs do not appear to have different structural characteristics between ctrls and autophagy induced cells.
B- Western blot expression of ATG5 and ATG7 should be perform in the same conditions.
We added ATG5 and ATG7 in the new Figure 1C.
Figure 5: C-Control IgG is missing.
Control IgG is indicated as - . We clarified it in the legend of the new Figure 6.
Round 2
Reviewer 1 Report (Previous Reviewer 3)
Comments and Suggestions for Authors
Third comments
Original article: Autophagy Promotes Enrichment of Raft Components within Extracellular Vesicles Secreted by Human 2FTGH Cells
Authors: Valeria Manganelli, Luciana Dini, Stefano Tacconi, Simone Dinarelli, Antonella Capozzi, Gloria Riitano, Serena Recalchi, Tuba Rana Caglar, Roberta Misasi, Maurizio Sorice and Tina Garofalo
I appreciate the effort of the authors to improve the manuscript.
The quality of the manuscript has undoubtedly increased, there is only one point in my opinion that has not been resolved and that is the use of a control cellular line, a non-transformed cell.
In the response to my comments 2 and 3. The authors send three bibliographic references that if I have found well do not demonstrate that the cellular process of autophagy is the same in controls (normal non-transformed cells- normal fibroblasts) as in the 2FTGH cell line. I leave it to the good judgment of the editor to consider my observation.
Manganelli V, Salvatori I, Costanzo M, Capozzi A, Caissutti D, Caterino M, Valle C, Ferri A, Sorice M, Ruoppolo M, Garofalo T, Misasi R. Overexpression of Neuroglobin Promotes Energy Metabolism and Autophagy Induction in Human Neuroblastoma SH-SY5Y Cells. Cells. 2021 Dec 2;10(12):3394. doi: 10.3390/cells10123394. PMID: 34943907; PMCID: PMC8699457.
Manganelli V, Capozzi A, Recalchi S, Riitano G, Mattei V, Longo A, Misasi R, Garofalo T, Sorice M. The Role of Cardiolipin as a Scaffold Mitochondrial Phospholipid in Autophagosome Formation: In Vitro Evidence. Biomolecules. 2021 Feb 5;11(2):222. doi: 10.3390/biom11020222. PMID: 33562550; PMCID: PMC7915802.
Kim HT, Goldberg AL. The deubiquitinating enzyme Usp14 allosterically inhibits multiple proteasomal activities and ubiquitin-independent proteolysis. J Biol Chem. 2017 Jun 9;292(23):9830-9839. doi: 10.1074/jbc.M116.763128. Epub 2017 Apr 17. PMID: 28416611; PMCID: PMC5465503.
Author Response
I appreciate the effort of the authors to improve the manuscript.
The quality of the manuscript has undoubtedly increased, there is only one point in my opinion that has not been resolved and that is the use of a control cellular line, a non-transformed cell.
In the response to my comments 2 and 3. The authors send three bibliographic references that if I have found well do not demonstrate that the cellular process of autophagy is the same in controls (normal non-transformed cells- normal fibroblasts) as in the 2FTGH cell line. I leave it to the good judgment of the editor to consider my observation.
Manganelli V, Salvatori I, Costanzo M, Capozzi A, Caissutti D, Caterino M, Valle C, Ferri A, Sorice M, Ruoppolo M, Garofalo T, Misasi R. Overexpression of Neuroglobin Promotes Energy Metabolism and Autophagy Induction in Human Neuroblastoma SH-SY5Y Cells. Cells. 2021 Dec 2;10(12):3394. doi: 10.3390/cells10123394. PMID: 34943907; PMCID: PMC8699457.
Manganelli V, Capozzi A, Recalchi S, Riitano G, Mattei V, Longo A, Misasi R, Garofalo T, Sorice M. The Role of Cardiolipin as a Scaffold Mitochondrial Phospholipid in Autophagosome Formation: In Vitro Evidence. Biomolecules. 2021 Feb 5;11(2):222. doi: 10.3390/biom11020222. PMID: 33562550; PMCID: PMC7915802.
Kim HT, Goldberg AL. The deubiquitinating enzyme Usp14 allosterically inhibits multiple proteasomal activities and ubiquitin-independent proteolysis. J Biol Chem. 2017 Jun 9;292(23):9830-9839. doi: 10.1074/jbc.M116.763128. Epub 2017 Apr 17. PMID: 28416611; PMCID: PMC5465503.
We thank the reviewer for appreciating the improvement of the manuscript according to his/her previous suggestions.
Regarding the question about the cell line, following our previous studies (Manganelli et al., Autophagy 2021, and Manganelli et al, Biomolecules 2021, we used the same cell line, untransfected control 2FTGH cells, in which it has been extensively shown that the cellular process of autophagy was not affected (as shown in Ref. 28, 29 and 38). Thus. the bibliographic references are:
- Manganelli, V.; Matarrese, P.; Antonioli, M.; Gambardella, L.; Vescovo, T.; Gretzmeier, C.; Longo, A.; Capozzi, A.; Recalchi, S.; Riitano, G.; Misasi, R.; Dengjel, J.; Malorni, W.; Fimia, G. M.; Sorice, M.; Garofalo, T. Raft-like lipid microdomains drive autophagy initiation via AMBRA1-ERLIN1 molecular association within MAMs. Autophagy 2021. 17, 2528-2548. doi: 10.1080/15548627.2020.1834207.
- Goldberg, A.A.; Nkengfac, B.; Sanchez, A.M.J.; Moroz, N.; Qureshi, S.T.; Koromilas, A.E.; Wang, S.; Burelle, Y.; Hussain, S.N.; Kristof, A.S. Regulation of ULK1 Expression and Autophagy by STAT1. J Biol. Chem. 2017. 292, 1899-1909. doi: 10.1074/jbc.M116.771584.
- Manganelli, V.; Capozzi, A.; Recalchi, S.; Riitano, G.; Mattei, V.; Longo, A.; Misasi, R.; Garofalo, T.; Sorice, M. The Role of Cardiolipin as a Scaffold Mitochondrial Phospholipid in Autophagosome Formation: In Vitro Evidence. Biomolecules 2021. 11, 222. doi: 10.3390/biom11020222.
These papers showed that Human fibrosarcoma 2FTGH cells could be considered superimposable to normal fibroblasts regarding the autophagy studies. In particular, in Ref. 28, to determine autophagy induction and autophagy activity, the following assays were performed:
- Flow cytometry by using Cyto-ID Autophagy Detection Kit. The kit was optimized for detection of autophagy in live cells by flow cytometry. This assay provides a rapid, specific, and quantitative approach for monitoring autophagic activity at the cellular level by using a 488 nm- excitable probe that becomes fluorescent in vesicles produced during autophagy.
- Immunofluorescence microscopy and flow cytometry after cell staining with anti-LC3 and anti-p62/SQSTM1 antibodies and western blot analysis. As far as Cyto-ID was concerned, a significant increase in green fluorescence emission was observed, as evidenced by higher median fluorescence values in starved fibroblasts than in control cells, which indicates the formation of LC3 puncta. Again, Western blot analysis revealed an increase of LC3- II after cell starvation together with a significant decreased of p62/SQSTM1 compared to untreated cells.
These data demonstrated that the autophagic process is the same in 2FTGH cells as in normal non-transformed fibroblasts, extensively studied in Ref. 4 (Garofalo et al., Autophagy 2016).
In Ref. 29 2FTGH, as well as MEF cells, were used to study autophagy induction. The autophagy flux was checked again by Western blot, observing increased lipidated LC3B (LC3B-II) levels after treatment with bafilomycin.
In Ref. 38 LC3B I, LC3B II and ULK1 were evaluated in 2FTGH cells, demonstrating that the cellular process of autophagy was not affected. Indeed, autophagy induction was checked in control and HBSS-treated cells by Western blot analysis, using anti-LC3 or anti-p62/SQSTM1 antibodies. Western blot analysis revealed an increase of LC3-II after cell starvation, together with a significant decreased of p62, compared to untreated cells, which was also confirmed by densitometric analysis.
Reviewer 2 Report (Previous Reviewer 1)
Comments and Suggestions for Authors
Thanks to the authors who made the necessary modifications. However, they made modifications or added western blot images that cause doubt. Please clarify.
Figure 1C: Original images of the blots of ATG 7 and ATG 8 are not available. Please provide.
Figure 3C: There are inconsistencies between the original image from the ANXA 1 and the one presented in Figure 3c. Please provide original blot images with molecular weight marker.
Line: 262: EVs instead of SVs
Figure 5A. There are inconsistencies between the original image from the cd63 and the one presented in Figure 5A. Please provide all original blot images with molecular weight marker.
Figure 6. Please provide all original blot images.
Author Response
Thanks to the authors who made the necessary modifications. However, they made modifications or added western blot images that cause doubt. Please clarify.
We thank the reviewer; we added the original images in the uncropped bot file and carefully checked MW markers in the Figures, adding new information in the Figures’ legends. In this concern, to avoid doubts, we show the complete image of control blot of CD63 in the new Supplementary Figure 1.
Figure 1C: Original images of the blots of ATG 7 and ATG 8 are not available. Please provide.
We added in the new file of uncropped blots the original images of the blots of the new Fig. 1C.
Figure 3C: There are inconsistencies between the original image from the ANXA 1 and the one presented in Figure 3c. Please provide original blot images with molecular weight marker.
We show in the new file of uncropped blots the original images of the blot of ANXA 1 with carefully checked molecular weight markers. According to the datasheet of the antibody the predicted MW is 35 KDa, although slight differences may be observed, depending on post translational modifications.
Line: 262: EVs instead of SVs
We corrected this word.
Figure 5A. There are inconsistencies between the original image from the cd63 and the one presented in Figure 5A. Please provide all original blot images with molecular weight marker.
We decided to show the original blot images of CD63 with the original molecular weight markers in the new Figure 5A.
We added in the Figure’s legend: “The observed MW may vary in the range of 25-65, *(predicted MW: 26 kDa), depending on post translational modifications, post translational cleavages, relative charges and other experimental factors, including unreducing condition. Therefore, there really is no consensus molecular weight”, as reported in the datasheet of anti-CD63 (abcam ab216130).
Figure 6. Please provide all original blot images.
We added in the new file of uncropped blots the original images of the blots of Fig. 6.

Reviewer 3 Report (New Reviewer)
Comments and Suggestions for Authors
The authors have addressed my concern in revised manuscript
Author Response
We thank the reviewer for his/her positive comment.
Round 3
Reviewer 2 Report (Previous Reviewer 1)
Comments and Suggestions for Authors
Accept in present form
This manuscript is a resubmission of an earlier submission. The following is a list of the peer review reports and author responses from that submission.
Round 1
Reviewer 1 Report
Comments and Suggestions for Authors
The authors present an interesting work where they report that autophagy promotes enrichment of raft components within EVs, however I have some comments:
Figure 1A. Do the magnified areas correspond to the source image? There are areas that do not coincide or cause doubts.
Figure 1B. There is no scale bar in the images.
Line 152: It is box instead of boxes.
Line 199: It is * 0.05 or ** 0.005, please clarify
Figure 3 B. The authors present a size distribution graph of both large vesicles and small vesicles. These data are taken from transmission electron microscopy images and they do not mention how many images they used to obtain said data. One of the requirements to characterize extracellular vesicles is to know size, concentration, mean, average, etc., and even more so in a comparative work, however, with tem images such data cannot be obtained, which is why it is necessary that the authors perform a nanoparticle analysis (NTA, DLS, etc.).
Line 540: No result is presented with a significant difference with a value of 0.0001 (****), please remove it.
Author Response
The authors present an interesting work where they report that autophagy promotes enrichment of raft components within EVs, however I have some comments:
We sincerely thank the Reviewer for appreciating our work.
Figure 1A. Do the magnified areas correspond to the source image? There are areas that do not coincide or cause doubts.
We thank the Reviewer for this comment which allowed us to organize Figure 1A in a more readable manner, that also allows to better compare the differences between the samples. The magnifications of the cell images are all identical. The third image of the first line in Figure 1A is not accompanied by a source image and serves to better show a MBV at higher magnification.
Figure 1B. There is no scale bar in the images.
We added scale bar.
Line 152: It is box instead of boxes.
Ok. We checked.
Line 199: It is * 0.05 or ** 0.005, please clarify
Ok, we clarified that it is 0.05.
Figure 3 B. The authors present a size distribution graph of both large vesicles and small vesicles. These data are taken from transmission electron microscopy images and they do not mention how many images they used to obtain said data. One of the requirements to characterize extracellular vesicles is to know size, concentration, mean, average, etc., and even more so in a comparative work, however, with tem images such data cannot be obtained, which is why it is necessary that the authors perform a nanoparticle analysis (NTA, DLS, etc.).
In Figure 3B the size distribution was performed by evaluating the diameter of vesicles present in five images TEM for each sample.
in our work the intent is to compare two high resolution microscopy techniques (TEM and AFM) for the sizing of isolated vesicles starting from the same sample preparation. This approach is useful for a comparison based solely on the resolution of the microscopy method used. NTA is not useful for our purposes since the sample is measured in solution in its native form (therefore a different starting sample). Indeed, several papers show the greater accuracy of TEM microscopy than using the NTA standard.
In particular:
- a work by E. Van der Pol et al., 2014 (DOI: 10.1111 / jth.12602) showed how, by measuring the size distribution of standard polystyrene nanobeads of different diameters, the relative error of the size and the coefficient of variation were the most variable when compared to other measurement techniques (i.e., TEM , RPS, conventional flow cytometry and dedicated flow cytometry).
- in another review by Bağcı et al., 2022, the different limitations of the NTA and how the technique distorts the detection of larger particles is well described.
Line 540: No result is presented with a significant difference with a value of 0.0001 (****), please remove it.
Ok, we removed the 0.0001 value.
Reviewer 2 Report
Comments and Suggestions for Authors
In this manuscript, the authors found autophagy could promote enrichment of raft components within EVs. This finding reveals the crosstalk between autophagy and endolysosomal system may have important implications in the understanding of cellular processes and potential therapeutic targets for diseases. While this is an interesting topic with new findings well demonstrated overall, the introduction and discussion of this field are not sufficient and some of the results raise concerns. I would like to recommend a major revision.
1. Did the authors observe or quantify the EV number difference between untreated and HBSS
2. It’s difficult to explain EV just by fraction number. Can the authors use specific EV markers and size measurements to identify large and small EV fractions isolated by iodixanol gradient density? For example, fractions 2-5 refer small EV with size around ?? nm and sEV markers (); while fractions 6-12 refer microvesicles (large EVs) with size around ?? nm and MV markers ().
3. Previous studies have indicated that autophagy can enhance the lysosomal degradation of EVs. However, under certain conditions, such as treatment with rapamycin (10.1002/advs.201801313), exposure to physical stress (doi.org/10.1002/advs.202302622) and nutrient deprivation, autophagy activation would alter EV production and potentially modify cargo compositions. It would be beneficial for the authors to reference these studies, compare them to their own findings, and provide a more detailed discussion within the manuscript.
4. Is this phenomenon general or exclusive to 2FTGH cells? The authors should comment on this question and engage in further discussion.
5. Figure 2: 197-200. Please check the P number.
6. Figure 4: What did NV indicate?
Comments on the Quality of English LanguageMinor editing of English language required
Author Response
In this manuscript, the authors found autophagy could promote enrichment of raft components within EVs. This finding reveals the crosstalk between autophagy and endolysosomal system may have important implications in the understanding of cellular processes and potential therapeutic targets for diseases. While this is an interesting topic with new findings well demonstrated overall, the introduction and discussion of this field are not sufficient and some of the results raise concerns. I would like to recommend a major revision.
- Did the authors observe or quantify the EV number difference between untreated and HBSS
We added a new Table 1 which analyses the concentration of small EVs measured by TEM reporting an increase of EVs in HBSS-treated respect to untreated cells (p <0.05).
- It’s difficult to explain EV just by fraction number. Can the authors use specific EV markers and size measurements to identify large and small EV fractions isolated by iodixanol gradient density? For example, fractions 2-5 refer small EV with size around ?? nm and sEV markers (); while fractions 6-12 refer microvesicles (large EVs) with size around ?? nm and MV markers ().
The high-resolution iodixanol density gradient was performed on P120 fraction, in which only vesicles with a diameter of 40–150 nm (small EVs) are present. This gradient separates small extracellular membrane vesicles bearing the hallmarks of sEVs/exosomes from non-vesicular components. Low-density fractions (3-5) are indicative of small EVs, whereas high-density fractions (8-12) are enriched in non-vesicular (NV) components (see the new legend of Figure 4). CD63 and CD81 were used as specific markers for small EVs; fibronectin as a specific marker of non-vesicular components.
- Previous studies have indicated that autophagy can enhance the lysosomal degradation of EVs. However, under certain conditions, such as treatment with rapamycin (10.1002/advs.201801313), exposure to physical stress (doi.org/10.1002/advs.202302622) and nutrient deprivation, autophagy activation would alter EV production and potentially modify cargo compositions. It would be beneficial for the authors to reference these studies, compare them to their own findings, and provide a more detailed discussion within the manuscript.
In the Discussion section we pointed out that previous studies have indicated that autophagy can enhance the lysosomal degradation of EVs. In particular, treatment with rapamycin, exposure to physical stress and nutrient deprivation, autophagy activation would alter EV production and potentially modify cargo compositions. The new References were added (new Ref. 37,38).
- Is this phenomenon general or exclusive to 2FTGH cells? The authors should comment on this question and engage in further discussion.
We suggest that this phenomenon is not exclusive to 2FTGH cells. This cell line represents a good model to study enrichment of raft components within EVs promoted by autophagy, since in previous studies (Ref. 23, 33) we studied in this cell line the role of raft components in autophagosome formation. We added few sentences in the Discussion section, indicating the limitation of this cell model.
- Figure 2: 197-200. Please check the P number.
p < 0.05.
- Figure 4: What did NV indicate?
NV indicate non vesicular bodies. We clarified it in the legend of Figure 4.
Reviewer 3 Report
Comments and Suggestions for Authors
Original article: Autophagy Promotes Enrichment of Raft Components within Extracellular Vesicles Secreted by Human 2FTGH Cells
Authors: Valeria Manganelli, Luciana Dini, Stefano Tacconi, Simone Dinarelli, Antonella Capozzi, Gloria Riitano, Serena Recalchi, Tuba Rana Caglar, Roberta Misasi, Maurizio Sorice and Tina Garofalo
The proposal of the article is interesting, the authors repeatedly argue the importance of using high resolution microscopes as a suitable strategy in the investigation of extracellular vesicle formation during autophagy. However, throughout the paper there are several inconsistencies that should be better addressed so that I have the opportunity to consider the work in a future publication.
Points for improvement are described in more detail below.
The in vitro model should justify, why use the human sarcoma cell line HT 1080. What is the rationale for using a tumorigenic line in autophagy. This cell line has several mutations and alterations specifically related to interferon signaling pathway. The authors should demonstrate or justify that modifications in the cell line do not alter the autophagy process or in the formation of extracellular vesicles under starvation studies. No description or origin of the cell line used was found throughout the manuscript. The context of the project should be modified to perform a study in a tumorigenic line and approached from the context of the microenvironment and essential metabolic needs required by the transformed cells, which in some sections are approached only as fibroblasts.
In the introduction, everything concerning autophagy and the formation of extracellular vesicles as key molecules is presented, so it is suggested to change the cell line of study to a non-transformed cell line or primary culture. If the approach is to identify a novel mechanism for multivesicular bodies in cancer, there should be a control, i.e., a primary culture or a non-transformed cell line, to determine that the MVBs phenomenon is characteristic of human fibrosarcoma.
Reiteratively the authors argue that they have improved methodologies for EVs that in reality there is no evidence of any improvement in the method used or obtaining efficient of EVs. On the contrary, Figure 4AC in the F5 fraction shows the heterogeneity of the sample. Throughout all the TEM images, different cell morphology is observed for the same treatment. That is, for example, in Figure 1A for the untreated group only one region shows protrusions and buds in the first image and for the rest of the micrographs of that group no protrusions and buds are observed.
A disadvantage of using Transmission Electron Microscopy and Atomic Force Microscopy (AFM) is the limitation of the fields analyzed. Due to the above and the inconsistency in the size and scale bar of the images, a count should be performed...i.e. in all treated cells the above mentioned changes were observed? When comparing treatments or samples the micrographs should be of the same size, at the same magnification with the same scale bar.
Images with inconsistencies in the scale bar or in the size of the image presented are shown below.
Figure 1-A, different magnification sizes and different bar sizes make it difficult to compare treatments.
Figure 1-B different scales on all axes.
Figure 2-A In the micrographs all the squares in the HBSS group are larger.
Figure 2-B Roughness and waviness were observed in all cells?. The deviation of the quantification even though it is at different scales practically caused very little data depertion.
Figure 3-A. The scale bar is of different sizes and the scale is not indicated in the image or in the text even though the micrographs are of the same size, the images for the HBSS treatment are always magnified to a larger scale.
In the introduction section they indicate that they characterized the molecular composition of autophagy induced EVs in human 2FTGH (2F) cells. However, no proteomic or metabolomic typing or determination of the content of the EVs was performed. The text should be modified and specify the molecules detected, so as not to suggest other types of studies.
Figure 1C. The WB image clearly shows the increase of LC3-II but not of LC3-I, and in fact the sum of the densitometry of the bands seems to be very similar in sum. Biologically, what does the increase or decrease of LC3 mean, depending on the case for I and II. The conclusion of the image they give is that there is a summation of pathways between autophagy and exosome biogenesis. However, clearly in the WB image a decrease in P62 is seen. Therefore, it can be concluded that starvation studies decrease MVB. What would be the biological explanation for the decrease in P62?. It is important to consider that the WB assays were performed with total cell lysates.
Figures 3-C and 4-A report different isoforms for CD63, what is the biological explanation for reporting different isoforms.
All images of spotted onto nitrocellulose strips should be performed in an electrophoretic run and send complete evidence of the membranes revealed in WB.
Author Response
The proposal of the article is interesting, the authors repeatedly argue the importance of using high resolution microscopes as a suitable strategy in the investigation of extracellular vesicle formation during autophagy. However, throughout the paper there are several inconsistencies that should be better addressed so that I have the opportunity to consider the work in a future publication.
Points for improvement are described in more detail below.
The in vitro model should justify, why use the human sarcoma cell line HT 1080. What is the rationale for using a tumorigenic line in autophagy.
We planned to analyze the role of autophagy on raft components within EVs in a tumorigenic cell line since in tumor cells EVs are known to have a significant impact on migration, invasion and metastasis (Ref. 17) as reported in Introduction section.
This cell line has several mutations and alterations specifically related to interferon signaling pathway. The authors should demonstrate or justify that modifications in the cell line do not alter the autophagy process or in the formation of extracellular vesicles under starvation studies.
This cell line is usually employed as a control line in several studies, including analysis of the interferon signaling pathway. In previous studies (Ref. 23 and new Ref. 24, 25), the autophagy process was analyzed in detail in these cells, demonstrating that it is not affected. We added this information in the Introduction section.
No description or origin of the cell line used was found throughout the manuscript.
The parent 2FTGH cells (provided by ECACC, 12021508) were derived from HT1080 human sarcoma cells. We added this information in Material and Methods section and added the new citation (Ref. 51).
The context of the project should be modified to perform a study in a tumorigenic line and approached from the context of the microenvironment and essential metabolic needs required by the transformed cells, which in some sections are approached only as fibroblasts.
We replaced fibroblasts with the correct fibrosarcoma cells.
In the introduction, everything concerning autophagy and the formation of extracellular vesicles as key molecules is presented, so it is suggested to change the cell line of study to a non-transformed cell line or primary culture. If the approach is to identify a novel mechanism for multivesicular bodies in cancer, there should be a control, i.e., a primary culture or a non-transformed cell line, to determine that the MVBs phenomenon is characteristic of human fibrosarcoma.
In this study we planned to analyze the role of autophagy on raft components within EVs only in a tumorigenic cell line since in tumor cells EVs are known to have a significant impact on migration, invasion and metastasis (Ref. 17). In particular, we used Human 2FTGH cells since in previous studies (Ref, 23, 33) we studied in this cell line the role of raft components in autophagosome formation.
Thus, in this manuscript our aim was to analyze whether rafts components are enriched within EVs cargoes, released from 2FTGH cells following autophagy triggering and not to identify a novel mechanism for multivesicular bodies in cancer.
Reiteratively the authors argue that they have improved methodologies for EVs that in reality there is no evidence of any improvement in the method used or obtaining efficient of EVs.
We agree with the reviewer that, although ultracentrifugation and iodixanol density gradient fractionation represent an improved methodology for EVs isolation, no further improvement of this technology is provided by the present manuscript. Thus, we modified the text accordingly.
On the contrary, Figure 4AC in the F5 fraction shows the heterogeneity of the sample. Throughout all the TEM images, different cell morphology is observed for the same treatment. That is, for example, in Figure 1A for the untreated group only one region shows protrusions and buds in the first image and for the rest of the micrographs of that group no protrusions and buds are observed.
A more suitable image was showed in the new Figure 4C.
We agree with the reviewer that Figure 1A does not prompt to state that several protrusions and buds on the cell surface were observed. We modified the text accordingly.
A disadvantage of using Transmission Electron Microscopy and Atomic Force Microscopy (AFM) is the limitation of the fields analyzed. Due to the above and the inconsistency in the size and scale bar of the images, a count should be performed...i.e. in all treated cells the above mentioned changes were observed?
We added the new Table 1 which analyses the concentration of small EVs measured by TEM reporting an increase of EVs in HBSS-treated respect to untreated cells (p <0.05).
When comparing treatments or samples the micrographs should be of the same size, at the same magnification with the same scale bar. Images with inconsistencies in the scale bar or in the size of the image presented are shown below.
Figure 1-A, different magnification sizes and different bar sizes make it difficult to compare treatments.
We agree with the reviewer's comment and the images are presented with uniform enlargements.
Figure 1-B different scales on all axes.
We agree with the reviewer's comment and the images in Figure 1B, now completely restructured, allow a simple reading of the morphological differences highlighted between untreated cells and HBSS cells.
Figure 2-A In the micrographs all the squares in the HBSS group are larger.
We amended the image by enlarging the images of untreated cells showed in panel A.
Figure 2-B Roughness and waviness were observed in all cells? The deviation of the quantification even though it is at different scales practically caused very little data depertion.
Roughness and waviness were measured on all the AFM images acquired and the graphs in Figure 2B show their statistical relevance and meaning. On this regard, we added distinct profiles for each parameter respectively for untreated and HBSS-treated cells. In particular, while the waviness depends on morphology of the cell and can be used to determine the overall arrangement of the nuclear vs perinuclear/cellular edge regions, the roughness depends only on the fine details (namely the rough) of the cellular membrane and allows to evaluate its arrangement (e.g. status or presence of vesicles) regardless of the cellular morphology (e.g. the height) beneath. Thus, by looking at the roughness profile alone, no differences are evident between cells showed; however, when all the roughness profiles were considered, a significantly difference among the specimens are evident, as shown in panel 2-C. We add further clarifications in the Figure caption.
Figure 3-A. The scale bar is of different sizes and the scale is not indicated in the image or in the text even though the micrographs are of the same size, the images for the HBSS treatment are always magnified to a larger scale.
We agree with the reviewer’s comment, so we have resized the images to make the magnifications uniform.
In the introduction section they indicate that they characterized the molecular composition of autophagy induced EVs in human 2FTGH (2F) cells. However, no proteomic or metabolomic typing or determination of the content of the EVs was performed. The text should be modified and specify the molecules detected, so as not to suggest other types of studies.
We agree with the reviewer and modified the text at the end of the Introduction section, indicating that, following our previous publications (Ref. 23, 33), we analyzed whether raft components are enriched in EVs after autophagy triggering.
Figure 1C. The WB image clearly shows the increase of LC3-II but not of LC3-I, and in fact the sum of the densitometry of the bands seems to be very similar in sum. Biologically, what does the increase or decrease of LC3 mean, depending on the case for I and II. The conclusion of the image they give is that there is a summation of pathways between autophagy and exosome biogenesis. However, clearly in the WB image a decrease in P62 is seen. Therefore, it can be concluded that starvation studies decrease MVB. What would be the biological explanation for the decrease in P62?. It is important to consider that the WB assays were performed with total cell lysates.
In Figure 1C we verified the autophagy activation after 4h cell starvation. Western blot analysis revealed an increase of LC3-II together a decrease of LC3-I. As expected, we observed a significant decrease of p62/SQSTM1 together with the increase of LC3-II after cell starvation, as also confirmed by densitometric analysis, demonstrating the integrity of the autophagic process in these cells. Concurrently, under our experimental conditions, the activation of the autophagic process is accompanied by heterogeneous MVB-like structures generation, as revealed by TEM. In this regard, previous biochemical studies support the evidence that autophagy shares with the molecular machinery of EVs, which include autophagy-related proteins and key proteins for EV biogenesis and secretion pathways (Ref. 2).
Figures 3-C and 4-A report different isoforms for CD63, what is the biological explanation for reporting different isoforms.
Under unreducing condition different isoforms for CD63 were detected, in agreement with several previous manuscripts (see Ref. 29).
All images of spotted onto nitrocellulose strips should be performed in an electrophoretic run and send complete evidence of the membranes revealed in WB.
In these experiments non proteic molecules were detected. Thus, in this case, dot blot analysis represents the most suitable methodological approach to detect ganglioside GD3. We agree with the reviewer that control LC3-II should be showed in WB analysis. We modified Figure 5C accordingly.
Round 2
Reviewer 1 Report
Comments and Suggestions for Authors
There are inconsistencies in the information that the authors provide to the reviewer with respect to the paper. In Figure 3B authors present a quantification of the size distribution of extracellular vesicles carried out by TEM, however, in the response to the reviewer they mention that 5 images were taken for each sample, while in the paper they mention that there were 12 images for each sample. Is there data manipulation?
Likewise, in the methodology, authors mention that the quantifications were carried out only with the images of the small vesicles, however, in the results (section 2.3.) mention that the quantifications were carried out on the P15 pellet (large vesicles) and the P120 pellet (small vesicles).
On the other hand, in section 2.3. CHARACTERIZATION OF EXTRACELLULAR VESICLES, I suggest the authors adhere to the MISEV guidelines (2018, 2024). The use of NTA is recommended for samples with particles between 40 and 400 nm, for what the authors present in their graphs, an analysis of nanoparticles can perfectly be carried out. Likewise, in section 2.3 "Characterization of small and large EVs secreted by human 2FTGH cells after the induction of autophagy" the authors mention: “To characterize the different subtypes of EVs secreted by human 2FTGH cells in the extracellular medium after the induction of autophagy, Conditioned medium was collected from untreated and HBSS-treated cells after 24 h of incubation in EV-depleted medium and subjected to standard differential centrifugation, as described above." It is understood that it is not the same sample.
I invite authors to make the appropriate changes before publishing the paper.
Reviewer 2 Report
Comments and Suggestions for Authors
The authors have satisfactorily addressed my concerns.
Reviewer 3 Report
Comments and Suggestions for Authors
I appreciate the effort of the authors to improve the manuscript.
There are still several points to be made in full or to answer and justify ones of the observations made.
Please check the file